# Instantaneous tracking of earthquake growth with elastogravity signals

Andrea Licciardi[1✉], Quentin Bletery[1], Bertrand Rouet-Leduc[2,3], Jean-Paul Ampuero[1] & Kévin Juhel[1,4]

Rapid and reliable estimation of large earthquake magnitude (above 8) is key to mitigating the risks associated with strong shaking and tsunamis[1]. Standard early warning systems based on seismic waves fail to rapidly estimate the size of such large earthquakes[2–5]. Geodesy-based approaches provide better estimations, but are also subject to large uncertainties and latency associated with the slowness of seismic waves. Recently discovered speed-of-light prompt elastogravity signals (PEGS) have raised hopes that these limitations may be overcome[6,7], but have not been tested for operational early warning. Here we show that PEGS can be used in real time to track earthquake growth instantaneously after the event reaches a certain magnitude. We develop a deep learning model that leverages the information carried by PEGS recorded by regional broadband seismometers in Japan before the arrival of seismic waves. After training on a database of synthetic waveforms augmented with empirical noise, we show that the algorithm can instantaneously track an earthquake source time function on real data. Our model unlocks 'true real-time' access to the rupture evolution of large earthquakes using a portion of seismograms that is routinely treated as noise, and can be immediately transformative for tsunami early warning.

The sudden displacement of rock mass induced by an earthquake generates density variations that, in turn, modify the Earth's gravity field. The signal associated with these transient gravity perturbations propagates at the speed of light, much faster than the fastest elastic waves (P-waves)[8–10]. Recent theoretical studies have shown the potential for earthquake early warning systems (EEWS) that are based on the gravity signals that would be measured by a future generation of gravity-gradient sensors[11,12], yet to be developed. In practice, existing inertial sensors (for example, seismometers) measure a combination of the direct gravity perturbations and their induced elastic response, named prompt elastogravity signals (PEGS)[13,14]. PEGS detection on real data is difficult for two reasons. First, the amplitude of the direct gravity perturbations is very small. Second, the induced elastic response tends to cancel out the gravity effects on seismometer recordings, especially in the early portion of the signal. The combination of these effects results in detectability limited to a time window preceding the P-wave arrival, which depends on epicentral distance (between a few seconds to a few tens of seconds), where PEGS reach their maximum amplitudes (a few nm s$^{-2}$ at most)[14,15]. Nevertheless, PEGS could prove beneficial for EEWS. First, they travel at the speed of light and might provide extra time for alert. Second, PEGS do not saturate, as opposed to P-waves recorded by near-field seismometers that may clip during large earthquakes. Finally, given the wavelength of the signal and the smoothness of the generated wavefield[14], the spatial complexity of the rupture does not

substantially affect PEGS amplitudes[15]. For this reason, PEGS dependence on earthquake magnitude, focal mechanism and source time function (STF) has the potential to improve early characterization of earthquake size and source parameters, under a simple point-source approximation[15,16]. In this work we show that PEGS can efficiently be used to improve operational EEWS.

Given the expected level of background noise, previous works have suggested a limit for PEGS detectability to earthquakes with magnitude ($M_w$) above 8 (refs. [15,16]). The occurrence of $M_w > 8$ earthquakes poses a difficult challenge for conventional EEWS. On the one hand, subduction megathrust earthquakes require accurate and fast estimates of final magnitude to mitigate the risk associated with strong shaking and to forecast the potential size of tsunami waves[17–19]. On the other hand, EEWS based on point-source algorithms that rely on the first few seconds of P-waves tend to produce saturated magnitude estimation for such large earthquakes. One reason is that instruments saturate for very large events. Another more fundamental reason is that the early portion of seismograms simply does not contain enough information to distinguish between small and large earthquakes (which have longer duration) at the very early stage of rupture[5]. An example of this paradigm is the performance of the EEWS of the Japan Meteorological Agency (JMA) during the 2011 $M_w = 9.0$ Tohoku-Oki earthquake, which underestimated the final $M_w$ of the event to around 8.1 (ref. [20]). Although the deterministic nature of earthquake rupture is still debated, a growing amount of evidence

[1]Université Côte d'Azur, IRD, CNRS, Observatoire de la Côte d'Azur, Géoazur, Sophia Antipolis, France. [2]Disaster Prevention Research Institute, Kyoto University, Kyoto, Japan. [3]Los Alamos National Laboratory, Geophysics Group, Los Alamos, NM, USA. [4]Laboratoire de Planétologie et Géodynamique, UMR 6112, Nantes University, CNRS, Nantes, France. ✉e-mail: andrea.licciardi@geoazur.unice.fr

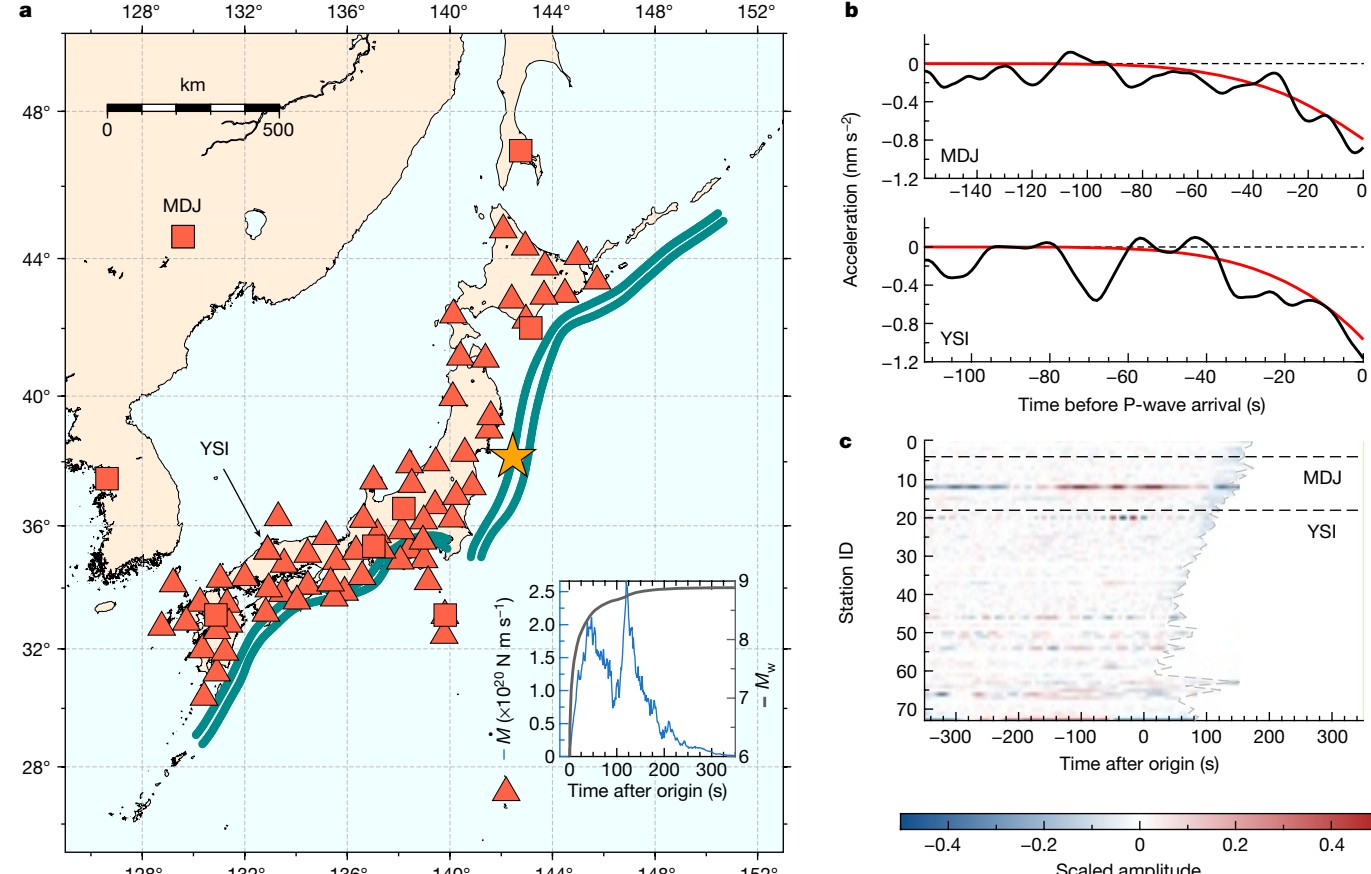

**Fig. 1 | Experimental set-up and input data examples from the synthetic database. a**, Locations of the stations in our network are indicated by light red symbols. Triangles are stations belonging to the F-Net seismic network and squares are stations from various networks obtained through the IRIS data centre. Dark green lines are made of 1,400 discrete locations of the synthetic earthquakes considered in our training database (see text). The orange star indicates the source location for one random synthetic example ($M_w$ = 8.9) extracted from the database whose moment rate (blue) and $M_w$ evolution (grey) are shown in the inset. **b**, The corresponding vertical waveforms at stations MDJ and YSI, where the red trace is the noise-free synthetic PEGS and the black trace is the same, with empirical noise added. Only the pre-P-wave time window is shown. **c**, Synthetic vertical waveforms plus noise associated with the selected event for each station in the seismic network. The grey dashed line indicates the P-wave arrival time. Stations are sorted by their longitude. The positions of MDJ and YSI are indicated by black horizontal dashed lines.

suggests that earthquake ruptures are not deterministic[21]. Therefore, EEWS should be designed to track the moment release as the rupture unfolds instead of forecast the final earthquake magnitude[3,22]. Over the last decade, finite-fault EEWS based on global navigation satellite system (GNSS) data have emerged as a new tool with which to overcome the magnitude saturation problem[23–28]. Nevertheless, subjective choices required in GNSS data selection and/or preprocessing[24,25] may result in large uncertainties. Moreover, the fast responses achieved for megathrust earthquakes[26] have recently been questioned and attributed not to the predictive power of GNSS data—which would enable the estimation of an earthquake's final magnitude before the rupture is over—but to prior constraints and regularization-induced artefacts[22]. A deep learning model based on GNNS data has recently been proposed to overcome these limitations[29]. Although it proved promising, as for other finite-fault approaches, it requires a priori assumptions on slip distribution. Finally, all existing EEWS suffer from unavoidable latency, owing to the speed at which the information is carried by P-waves, and therefore produce a time-shifted version of the earthquake STF.

In this context, we show that a convolutional neural network (CNN)[30] approach can leverage the information carried by PEGS at the speed of light to overcome these limitations for large earthquakes. Successful applications of deep learning in seismology have provided new tools for pushing the detection limit of small seismic signals[31,32] and for the

characterization of earthquake source parameters (magnitude and location)[33–35] with EEWS applications[29,36,37]. Here we present a deep learning model, PEGSNet, trained to estimate earthquake location and track the time-dependent magnitude, $M_w(t)$, from PEGS data before P-wave arrivals.

## The training database

We focus on the Japanese subduction zone because: (1) it has experienced one of the largest and most destructive subduction events (the 2011 $M_w$ = 9.0 Tohoku-Oki earthquake), and (2) we can rely on a large and dense network of good-quality broadband seismic stations (Fig. 1a).

Because of the paucity of PEGS observations[16], we train PEGSNet on a database of synthetic PEGS waveforms generated with a normal-mode summation code[14] and add real noise recorded at each station in the seismic network. The training database is made of 500,000 synthetic earthquake sources distributed along the Japanese megathrust, with location, strike and dip angles following the Slab2.0 model[38]. We draw random magnitude and rake angles following uniform (from 5.5 to 10) and Gaussian ($\mu$ = 90°, $\sigma$ = 10°) distributions, respectively. In this work, we consider only the most simplistic rupture spatial descriptions: point sources. The difference in PEGS predictions associated with the source finiteness was shown to be within data uncertainties

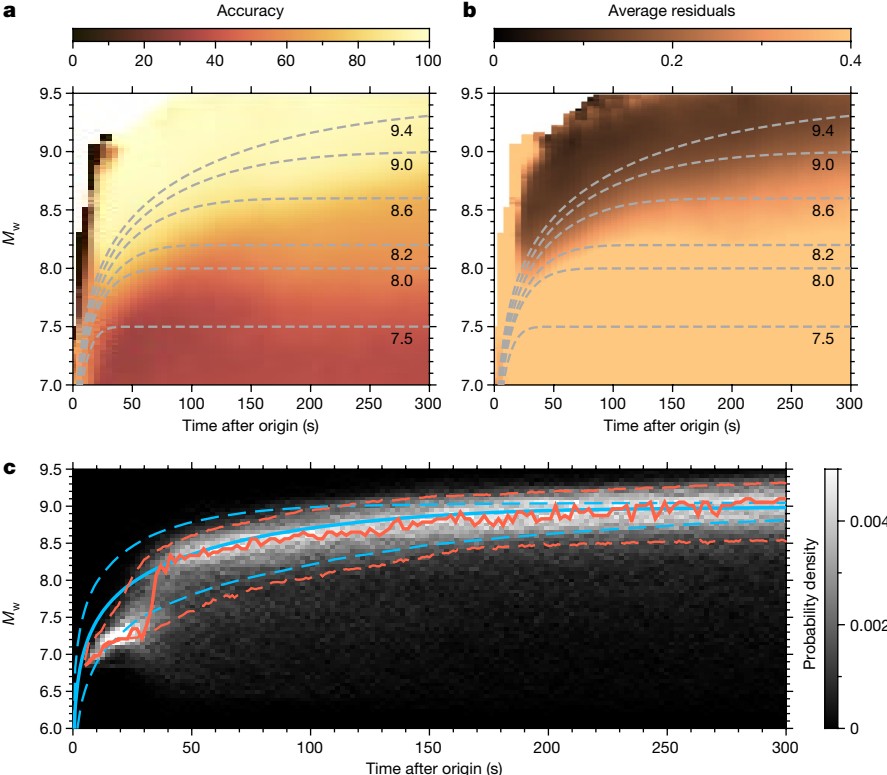

**Fig. 2 | Tracking $M_w(t)$ from PEGS data.** Results of predictions on the test set. **a**, Predictions accuracy as a function of time and $M_w$. For each pixel in the image, accuracy is calculated as the number of successful predictions divided by the total number of samples. A prediction is defined as successful if the distance with its corresponding label is within ±0.4 magnitude units. Dashed lines indicate the average values of the true $M_w(t)$ for events with different final magnitudes. **b**, As in **a**, but showing the average residuals ($|M_{w[true]} - M_{w[pred]}|$) for each pixel. **c**, Probability density of $M_w$ predictions for all the events in the test set with true final $M_w$ of 9 ± 0.05. The solid red line is the mode of the distribution. The red dashed lines bound the 25th–95th percentile range. The blue lines are the median (solid) and the 5th–95th percentile range (dashed) of the labels.

during the 2011 Tohoku earthquake[15]. The validity of this assumption is further corroborated by the detection of PEGS generated by multiple events considering a simple point-source model[16]. However, simple triangular STFs might not suffice for modelling PEGS for realistic earthquake ruptures[16]. For this reason, we generate random STFs using a model designed to mimic empirical laws and statistical observations[5] (Fig. 1a, inset, and Extended Data Fig. 1a) to produce three-component (east, north, vertical) waveforms representing the source characteristics of all expected large megathrust earthquakes. In this STF model the smaller and larger earthquakes start in the same way. We then add empirical noise recorded at each station (Fig. 1b) to the generated waveforms. Both synthetics and noise seismograms are decimated to 1 Hz and bandpass-filtered between 2 mHz (high-pass Butterworth, two poles, causal) and 30 mHz (low-pass Butterworth, six poles, causal), to enhance long-period PEGS amplitudes by suppressing higher-frequency noise[7]. Traces are clipped to a threshold value of ±10 nm s$^{-2}$ (the maximum expected amplitude for PEGS in our database) and scaled by the same value to facilitate convergence of the optimizer, and at the same time to preserve information about PEGS radiation patterns across the seismic network. Finally, we store each example as 700-s-long traces centred at the origin time of the earthquake (Fig. 1c), with its three labels corresponding to the latitude ($\varphi$), longitude ($\lambda$) and the time-integrated STF of the event converted to the time-dependent magnitude $M_w(t)$.

## Building PEGSNet, labelling and training

PEGSNet is a deep CNN that combines convolutional layers and fully connected layers in sequence (Extended Data Fig. 2; details

in Methods). There exists a wealth of literature on deep learning architectures, but our goal here is to show the feasibility of PEGS-based EEWS using deep learning, and we therefore use a simple and classic Le-NET-like CNN architecture[39]. Similar architectures have been used for magnitude and location estimation using P and S-waves with a single-station approach to increase the number of training examples[33,34]. Here we take advantage of our large synthetic database and adopt a multi-station approach. CNNs work best with image-like inputs, and we arrange our data as an image, sorting stations by their longitude (Fig. 1c).

Our aim is to track the time-dependent magnitude, $M_w(t)$, which corresponds to the time-integrated STF at a given time. For PEGSNet to learn $M_w(t)$, we design a specific strategy: we randomly select the starting time ($T_1$) of the analysed data during training. For each input example, we extract 315 s of data from $T_1$ to $T_2 = T_1 + 315$ s and assign the value of $M_w(T_2)$ as the corresponding label. This allows the model to learn patterns in the data as the STF evolves with time.

We exclusively use the information carried by PEGS by setting to zero the amplitude of the seismograms for $t \geq T_P$, where $T_P$ is the P-wave arrival time at each station for a given event. Therefore, P-wave arrival times are assumed to be known—in practice, any existing triggering algorithms (whether based on deep learning or not) could be used to obtain them before feeding the data to the model. The final input data for each example consists of a three-channel image (one for each seismic component), and the output layer of PEGSNet predicts three real values for $M_w(T_2)$, $\varphi$ and $\lambda$.

The synthetic database described above is randomly split into 70% training, 20% validation and 10% test sets, with a given synthetic STF and a given empirical noise measurement going to only one of the datasets.

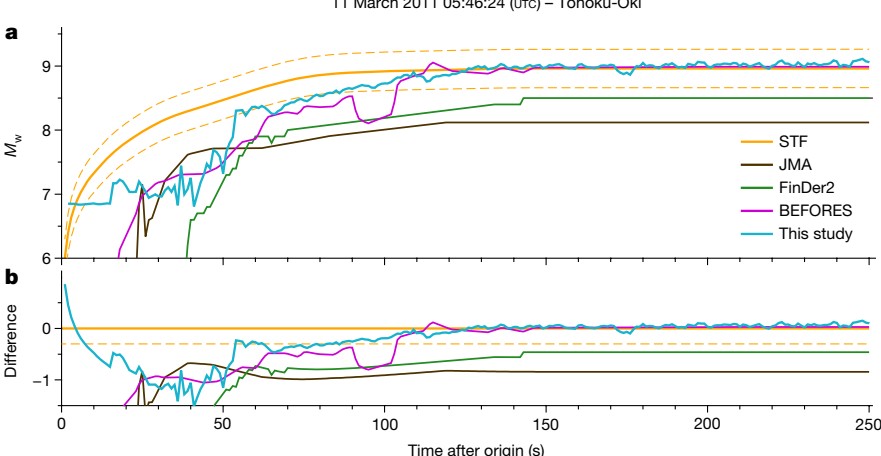

**Fig. 3 | Moment tracking of the 2011 $M_w$ 9.0 Tohoku-Oki earthquake. a**, Comparison of PEGSNet $M_w(t)$ predictions (blue) with the source time function (STF) of the Tohoku-Oki earthquake[40] (orange) and with the results of three existing EEWS (JMA[20], FinDer2[41] and BEFORES[24]). Dashed orange lines indicate ±0.3 magnitude units. **b**, Difference between the predicted $M_w$ of various algorithms (PEGSNet in blue) and the 'true' STF as shown in **a**. The solid orange line indicates a difference of zero, the dashed orange line a difference of −0.3 $M_w$ units (underestimation).

Training is performed by minimizing the Huber loss between the labels (the true values of $M_w(t)$, $\varphi$ and $\lambda$ are known) and the predicted values. The model with the lowest error on the validation set is chosen as the final model.

## Instantaneous tracking of $M_w$

Figure 2 summarizes the results on the synthetic test set for $M_w(t)$, and those for $\varphi$ and $\lambda$ are displayed in Extended Data Fig. 3. Results for location rely on the P-wave arrival times (which are assumed to be known) and show errors around 25 to 30 km starting at about 50 s after origin. In the following, we will exclusively discuss the results for $M_w(t)$. These results represent the main contribution of this work as they are obtained using only the pre P-wave portion of the seismograms, which is routinely treated as pure noise. For each example in the test set, we simulate a real-time scenario by parsing the data with a running time window of 315 s. Each data window is fed to PEGSNet, which makes an $M_w$ estimate for the end of the window, and the window is shifted in steps of 1 s as PEGSNet attempts to progressively reconstruct the STF (Extended Data Fig. 4). We define a successful prediction if the estimated $M_w(t)$ lies within ±0.4 magnitude units from the true value.

PEGSNet can track the moment released by earthquakes with final $M_w$ above 8.6 with good accuracy (above 90%, Fig. 2a) and low errors ($M_w$ error below 0.25, Fig. 2b), starting at about 40 s after origin time. For earthquakes with final $M_w$ between 8.2 and 8.6, early tracking of the moment release is more difficult and only the final $M_w$ can be estimated (with accuracy between 60% and 70% and errors above 0.25) after 150 s from origin time. Any predicted $M_w$ below 8.2 is poorly constrained by the data and we place a conservative lower limit on PEGSNet sensitivity to $M_w$ at 8.3. However, this limit depends on the noise amplitudes across the seismic network. Under favourable noise conditions (standard deviation of the noise for the whole seismic network below around 0.5 nm s$^{-2}$), it can be reduced to about 7.9 or 8.0 (Extended Data Fig. 5).

To examine the time-dependent performance of PEGSNet more closely, Fig. 2c shows all the predictions (on the test set) associated with events with a final $M_w$ of 9.0 ± 0.05. In the first 30–40 s after origin, predictions are strongly underestimated. This arises from a combination of two effects: the lack of sensitivity below 8.3 (in the first 30–40 s after origin the current magnitude is below 8.3 in our STF database) and the fact that, at stations located close to the source,

P-waves can mask PEGS, owing to their small time separation. A test performed on noise-free synthetics shows that at least 15 s after origin are needed by the model to reliably predict $M_w$ (Extended Data Fig. 6). Starting at about 40 s (when the true $M_w$ is generally above 8.2 in our STF database), PEGSNet can track the evolution of the moment release instantaneously, that is, without time shift between estimated and true $M_w(t)$, as indicated by the mode of the distribution of the predictions. Our results show that PEGSNet can exploit the key features of PEGS: (1) PEGS sensitivity to magnitude for large earthquakes allows the model to distinguish, for example, between an $M_w = 8$ and an $M_w = 9$ earthquake and (2) the information about $M_w$ is effectively propagated at the speed of light, which results in instantaneous tracking of the moment release, from about 40 s into the rupture until its end. Although P-wave triggering is needed at each station, the instantaneous information about the source carried by PEGS is readily available in the data that precede P-wave arrivals. This allows PEGSNet to estimate $M_w(t)$ with zero delay once the magnitude exceeds approximately 8.3.

## Playback of the Tohoku-Oki earthquake

We tested PEGSNet on real data from the 2011 $M_w$ 9.0 Tohoku-Oki earthquake. The raw seismograms are processed as described in Methods and fed to PEGSNet. We simulated a data playback scenario and made an $M_w$ prediction at each second starting at the earthquake origin time and using only the previous 315 s of data (details in Methods). Figure 3 shows the results of this retrospective analysis compared with a 'true' STF calibrated against various different data types[40], the historical JMA EEWS performance[20], and the best-performing finite-fault EEWS based on seismic (FinDer2)[41] and GNSS data (BEFORES)[24]. Further comparisons with additional algorithms are shown in Extended Data Fig. 7. After around 55 s, PEGSNet is always closer to the 'true' STF than the other algorithms (Fig. 3a). The predicted $M_w$ between 55 and 100 s is underestimated by about 0.3 magnitude units compared to the ground truth. PEGSNet reaches a correct prediction of the final $M_w$ after about 120 s, when the rupture is almost over. Despite the slight underprediction, the predicted $M_w(t)$ values between 55 and 100 s show a clear increasing trend and suggest that the rupture is still in progress. Therefore, we argue that PEGS can be used in real time to track the evolving rupture as it unfolds. Consistently with our synthetic tests (Fig. 2), our predictions after about 55 s (corresponding to the time the earthquake

reaches $M_w \approx 8.3$) provide a snapshot of the evolution of the earthquake rupture at that exact time because PEGS carry information about the source at the speed of light. This is further corroborated by a test on synthetic data generated using the 'true' Tohoku-Oki earthquake STF and noise seismograms extracted from the test set (Extended Data Fig. 8).

Notably, PEGSNet does not suffer from the magnitude saturation issue that instead affects the seismic-based EEWS. Both in terms of latency and accuracy, the performance of the GNSS-based algorithm BEFORES is the closest to PEGSNet. However, the response of PEGSNet is faster: for an arbitrary threshold at $M_w = 8.0$, PEGSNet would issue an alert 53 s after origin, BEFORES about 8 s later. PEGSNet is more accurate: the difference with the 'true' STF is always equal to or less than $0.3\,M_w$ units starting at 55 s, whereas BEFORES reaches that level of precision only after 100 s from origin (Fig. 3b). More importantly, in contrast to GNSS-based EEWS, PEGSNet does not require prior assumptions on the data, because it is trained to learn features from the entire network of stations regardless of their data quality. In addition, no-slip distributions are assumed or modelled in our approach as PEGS are only sensitive to smooth variations of the released moment and, therefore, less likely to be influenced by the spatial complexity of the rupture.

To further assess the robustness of PEGSNet predictions on the Tohoku data, we performed a test in which we keep the P-wave arrival information but substitute the actual recorded waveforms in the pre-P-wave time window with noise. The resulting prediction of $M_w(t)$ never exceeds the lower sensitivity limit of PEGSNet (that is, 8.3) and remains constant at about 6.5, which provides a baseline value for noise (Extended Data Fig. 9). We also tested PEGSNet on all the subduction earthquakes (dip-slip mechanism within 40 km from the megathrust) with $M_w \geq 7$ that have occurred since January 2003, without considering aftershocks (Methods and Extended Data Fig. 10). For $M_w < 8$ earthquakes, PEGSNet predictions converge toward the noise baseline, confirming that these events are essentially not distinguishable from noise.

## Implications for early warning

We have demonstrated instantaneous tracking of moment release for large earthquakes (Fig. 3). Our results promote PEGS as a new class of observables, easily accessible from the recordings of currently deployed broadband seismometers worldwide, for practical application in early warning systems that are currently limited by the speed of P-waves. In the context of EEWS, PEGSNet can complement any existing algorithm (either seismic- or GNSS-based), to improve $M_w$ latency estimation and accuracy for $M_w > 8.3$ megathrust earthquakes. For example, PEGSNet could be combined with a recent deep learning model based on GNSS data[29] to eliminate intrinsic latency due to P-wave speed.

At the same time, PEGSNet can immediately prove critical for tsunami early warning for which $M_w$ estimation within a few minutes is vital. Continuous updates of current $M_w$ can be fed to predictive models of tsunami waves height, helping mitigate the associated risk.

Our results suggest that PEGS can play a key part in the early characterization of rupture evolution for large earthquakes. For such events, PEGS data represent a new and independent source of information to constrain the magnitude in real time.

PEGSNet requires only a few modifications to be implemented in real time. Once trained, PEGSNet predictions are quasi-instantaneous, although some latency could be introduced by the preprocessing step. Although tailored here to earthquakes from the Japanese subduction zone, PEGSNet can be easily adapted to other seismic networks and regions and source mechanisms. In particular, PEGSNet portability to other regions only requires the availability of noise recordings for the seismic network of interest.

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

# Methods

## Construction of the training database

We train PEGSNet on a database of synthetic data augmented with empirical noise. For each example in the database, three-component waveforms at each station location are obtained as follows.

**Source parameters.** The event source location is randomly picked from 1,400 possible locations extracted from the Slab2.0 model of subduction geometry[38] at two different depths (20 and 30 km). Given latitude, longitude and depth, strike and dip angles are determined by the subduction geometry and rake is randomly extracted from a normal distribution with mean = 90° and standard deviation of 10°. The event final $M_w$ is drawn from a uniform distribution with min = 5.5 and max = 10.0. We deliberately choose not to use a Gutenberg–Richter distribution for $M_w$ to avoid a sampling bias during training, by which the model might better estimate certain magnitude values simply because they are more represented in the training database. Finally, from $M_w$ we compute the scalar moment $M_0 = 10^{1.5M_w+9.1}$.

**Source time function.** Given $M_0$, a pseudo-empirical STF is computed using the STF model described in a previous work[5], which includes a multiplicative error term and is valid for earthquakes with $M_w > 7.0$. In summary,

$$\text{STF}(t) = M_0 \frac{f(t)}{\int f(t)\mathrm{d}t}, \tag{1}$$

with:

$$f(t) = t\exp\{-0.5(\lambda t)^2\}[1 + N(t)], \tag{2a}$$

$$\lambda = 10^{(7.24 - 0.41\log(M_0) + \varepsilon)}, \tag{2b}$$

$$N(t) = 0.38\frac{n(t)}{\sigma}, \tag{2c}$$

where $\varepsilon$ is drawn from a Gaussian distribution with zero mean and standard deviation of 0.15, $n(t)$ is the time integral of a Gaussian noise time series with zero mean and $\sigma$ is the standard deviation of $n(t)$. The term $\varepsilon$ accounts for variability in the STF duration for a given $M_0$, whereas $N(t)$ models the characteristics of noise observed in real STFs[5]. Examples of final STFs for different magnitude values are shown in Extended Data Fig. 1.

**Computing synthetic waveforms.** With the selected source parameters and STFs, we use the normal-mode approach described in a previous work[14] to compute three-component synthetic waveforms in a spatial domain of about 20° around the source epicentre. The resulting seismometer responses are convolved with the STF of the corresponding synthetic event and multiplied by the scalar moment to obtain synthetic traces of acceleration sampled at 1 Hz at each station location. Finally, traces are bandpass-filtered between 2.0 mHz (Butterworth, two poles, causal) and 30.0 mHz (Butterworth, six poles, causal). The final seismograms are 700 s long centred at the event origin time.

**Noise database.** The noise database consists of 259 days of three-component waveforms for two non-continuous time intervals: between January 2011 and October 2011 (excluding March 2011) and between January 2014 and April 2014. These intervals have been chosen to sample variable (seasonal) noise conditions. We note that the temporal range spanned by the noise database does not overlap with any of the earthquakes used for real data cases (Extended Data Fig. 10a). We first divide the daily recordings into 1-h-long traces and then apply limited preprocessing, removing the instrument response, the mean and the

linear trend, converting to acceleration and decimating the original traces from 20 to 1 Hz. Finally, each trace is filtered using the same bandpass filter applied to the synthetic seismograms (see previous step) and stored. Note that no a priori assumptions on levels and characteristics of the selected noise are made. On the contrary, we include all real noise conditions found in continuous seismic recordings in the specified period range. This is because, in principle, we want the model to be able to generalize well under a broad range of noise conditions.

**Adding empirical noise to synthetics.** From the noise database described in the previous step, a realization of noise (700 s long) is extracted by randomly selecting a starting time point. In this process, we make sure to use different noise data in training, validation and test sets. To preserve spatial coherence of noise across the seismic network, the same time interval is used for all stations for a given event. The selected noise traces are then added to the corresponding acceleration seismograms to produce the final input data for PEGSNet. If noise data are not available for one or more stations in the selected time window for a given event, we discard those stations by setting the corresponding final trace amplitudes (noise and PEGS) to zero in the input data.

**Preprocessing of input data.** Before being fed to PEGSNet, we first sort the input waveform for each example based on station longitude. We found this approach to be effective, but we note that the problem of concatenating station waveforms in a meaningful way in deep learning is an active area of research[35]. Then, on the basis of the theoretical P-wave arrival time ($T_P$) at each station for a given event, we set the amplitude of the seismograms to zero for $t \geq T_P$. Note that PEGSNet does not perform P-wave triggering itself. Instead, it relies on theoretical P-wave arrivals. In a real-time scenario, any existing P-wave triggering algorithm (whether based on machine learning or not) can be used to set the switch on the corresponding stations whose data can then be passed to PEGSNet.

To limit the influence of very noisy traces and to suppress high amplitudes (possibly related to the background regional seismicity), we further clipped the resulting trace using a threshold of ±10 nm s$^{-2}$. This threshold is chosen according to the maximum PEGS amplitude for an $M_w = 10$ earthquake at 315 s as found in the calculated database of noise-free synthetics. Amplitudes are finally scaled by 10 nm s$^{-2}$ to facilitate convergence of the optimizer, and at the same time, to preserve information about the relative amplitudes of the PEGS radiation pattern across the seismic network. Finally, to simulate missing data and/or problematic sensors, we randomly mute 5% of the stations for each event by setting to zero the amplitudes of the corresponding traces.

## Deep learning and PEGSNet

**Previous work.** Convolutional neural networks (CNNs) originated in neocognitron[42] and became practical once it was found that the backpropagation procedure[43] can be used to compute the gradient of an objective function with respect to the weights of the network. CNNs are a regularized form of neural networks, that is, the function space they represent is simpler and they are more sample-efficient than fully connected neural networks[44]. Deep CNNs have brought about a revolution in computer vision, and have had a role in almost every state-of-the-art approach for tasks related to recognition and detection in images[45,46]. In geoscience, machine learning has shown strong potential for data-driven discovery of previously unknown signals and physical processes hidden in large volumes of noisy data[47,48].

We note, however, that our choice of a deep learning model over classical machine learning models offers an appealing framework to directly deal with raw seismograms. As a consequence, this choice enables us to explore a larger function space that is not limited by building an a priori set of features, which is a requirement for applying classical machine learning models on seismogram data.

Successful applications of deep learning in seismology have provided new tools for pushing the detection limit of small seismic signals[31,32] and for the characterization of earthquake source parameters (magnitude and location)[33-35] with EEWS applications[29,36,37]. We present a deep learning model, PEGSNet, trained to estimate earthquake location and track the time-dependent magnitude, $M_w(t)$, from PEGS data, before P-wave arrivals.

**Description of PEGSNet architecture.** PEGSNet is a deep CNN that combines convolutional layers and fully connected layers in sequence (Extended Data Fig. 2a). The input of the network is a multi-channel image of size $(M, N, c)$ where $M$ is 315 (corresponding to 315-s-long traces sampled at 1 Hz), $N$ is the number of stations (74) and $c$ is the number of seismogram components used (three: east, north and vertical). The outputs of the network are three values corresponding to moment magnitude ($M_w$), latitude ($\varphi$) and longitude ($\lambda$), where $M_w$ is time dependent. The training strategy used to learn $M_w(t)$ from the data is described below.

The first part of the model (the CNN) consists of eight convolutional blocks. Each block is made of one convolutional layer with a rectified linear unit (ReLU) activation function followed by a dropout layer. The number of filters in each convolutional layer increases from 32 (blocks 1–5) to 64 (blocks 6–7) to 128 (block 8) to progressively extract more detailed features of the input data. A fixed kernel size of $3 \times 3$ is used in each convolutional layer. We use spatial dropout with a fixed rate of 4% to reduce overfitting of the training set. Maximum pooling layers are added starting from block 4 to reduce the overall dimension of the input features by a factor of 4. The output of the CNN is then flattened and fed to a sequence of two dense layers of size 512, and 256 with a ReLU activation function and standard dropout with a 4% rate. Fully connected layers perform the high-level reasoning and map the learned features to the desired outputs. The output layer consists of three neurons that perform regression through a hyperbolic tangent activation function (tanh). The labelling strategy for $M_w(t)$, $\varphi$ and $\lambda$ is discussed in detail below. The total number of parameters in the network is 1,479,427.

**Learning strategy.** The purpose of the model is to track the moment released by a given earthquake as it evolves from the origin time. A specific learning strategy has been developed to address this task (Extended Data Fig. 2).

*Labelling.* Labels are $\varphi$, $\lambda$ and a time-dependent $M_w$. $\varphi$, $\lambda$ simply correspond to the true values for each event. $M_w(t)$ is the time integration of the STF for each event. As detailed in the next section, the model is trained by randomly perturbing the ending time of the input seismograms, so that for a given ending time the input data are associated with the value of $M_w(t)$ at that time. To enforce the role of the tanh activation function in the output layer, we further scale all the labels to fall in the [−1, 1] interval through min/max normalization.

*Learning the time-dependent moment release.* In order for PEGSNet to learn $M_w(t)$, we randomly perturb the starting time of the input data during training (Extended Data Fig. 2c). Every time that an example is extracted from the dataset, a value ($T_1$) is drawn at random from a uniform distribution between −315 and 0 (s). $T_1$ is the time relative to the earthquake origin time ($T_0$) corresponding to the starting time of the selected seismograms for that example. In practice, from the 700-s-long seismograms (centred on $T_0$) in the database, we extract traces from $T_1$ to $T_2 = T_1 + 315$ s: for $T_1 = -315$ s the extracted traces end at $T_0$; for $T_1 = 0$ s the traces start at $T_0$ and end 315 s after. Once a value for $T_1$ is selected, the value of $M_w(T_2)$ is assigned as the corresponding label for this example (Extended Data Fig. 2d). This enables the model to learn patterns in the data as the STF evolves with time.

*Training.* The full database (500,000 examples of synthetic earthquakes) is split into training (350,000) validation (100,000) and test (50,000) sets, following a 70/20/10 strategy. The network is trained for 200 epochs (using batches of size 512) on the training set by minimizing the Huber loss between the true values and the predicted earthquake source parameters using the Adam algorithm[49], with its default parameters ($\beta_1 = 0.9$ and $\beta_2 = 0.999$) and a learning rate of 0.001. At the end of each epoch, the model is tested on the validation set to assess the learning performance and avoid overfitting (Extended Data Fig. 2b). After learning, the model that achieved the best performance (lowest loss value) on the validation set is selected as the final model. The final model is then tested against the test set (therefore with data that has never been seen by PEGSNet during training) to assess its final performance.

**Testing strategy.** Once PEGSNet is trained, it can be used to estimate $M_w(t)$ in a real-time scenario. We assess the latency performance of PEGSNet on the test set with the following procedure (Extended Data Fig. 4). For each example in the test set, we slide a 315-s-long window [$T_1, T_2 = T_1 + 315$ s] through the data with a time step of 1 s. The starting window ends at the earthquake origin time $T_0$ ($T_2 = T_0$ and $T_1 = T_0 - 315$ s) and the final window starts at the earthquake origin time ($T_2 = T_0 + 315$ s and $T_1 = T_0$). We let PEGSNet predict $M_w(T_2)$ at each time step, thus progressively reconstructing the STF. Each $M_w(t)$ estimate made by PEGSNet only makes use of information in the past 315 s. The same procedure is also applied to real data (Fig. 3 and Extended Data Fig. 10), to simulate a playback of the data as if they were fed to PEGSNet in real-time.

**Test on noise-free synthetics.** We investigate the performance of PEGSNet by using the same database described above but without including noise in the input data. Training and testing on noise-free synthetic data provides an upper limit of PEGSNet performance. Although this experiment represents a virtually impossible scenario for real-world applications, the results can reveal inherent limitations of our model or in the input data. Extended Data Fig. 6a shows the accuracy map for the test set. As expected, the model is able to determine the final $M_w$ of the events with high accuracy and similar performance regardless of the actual $M_w$ of the event, except at early times. To look at the latency performance more in detail, Extended Data Fig. 6b shows the density plot of the residuals as a function of time for the whole noise-free test set. Errors are mostly confined within ±0.1 but are relatively higher in the first 10–15 s after origin. We relate this observation to a combination of two factors: first, in the first 15 s after origin, very small PEGS amplitudes are expected at very few stations in the seismic network, partially owing to the cancelling effect between the direct and induced terms. This can lead to a situation in which little information is present in the input images and the model ends up predicting the mean value of the labels at these early times. Second, the seismic network geometry may not be optimal for recording PEGS amplitudes in this time window. Finally, we note that similar behaviour is observed for the results obtained on the noisy database (Fig. 2a, b) but with a higher latency (30–40 s). This highlights the role of noise in degrading the optimal performance of PEGSNet.

## Preprocessing of real data

The preprocessing steps for real data closely follow the procedure detailed in previous work[7]. For each station and each component:

1. Select 1-h-long raw seismograms ending at the theoretical $T_P$ calculated using the source location from the United States Geological Survey (USGS) catalogue (Extended Data Fig. 10a);

2. Remove the mean;

3. Remove the instrument response and obtain acceleration signals;

4. Lowpass 30.0 mHz (Butterworth, six poles, causal);

5. Decimate to 1 Hz;

6. Highpass 2.0 mHz (Butterworth, two poles, causal);

7. Clip to ±10 nm s$^{-2}$ and scale by the same value;

8. Pad with zeros for $t \geq T_P$ and select 700-s-long trace centred on $T_0$.

This procedure is the same as that used to generate the synthetic database, except that here, traces need to be cut at the P-wave arrival first to avoid contamination of PEGS by the P-wave during instrument response removal.

To speed up our testing procedure (see Methods subsection 'Testing strategy'), the data are preprocessed once and then sliced into input for PEGSNet at each time step. In an online version of our model, this is unfeasible as all the preprocessing steps need to be applied each time that new packets of data are streamed in. We simulate the conditions of a real-time workflow on the Tohoku-Oki data to assess potential discrepancies with the simplified workflow in the results: at each time step, we apply the preprocessing steps described above, using 1-h-long trace ending at the current time step. We find that the resulting PEGSNet predictions obtained using the two workflows are essentially indistinguishable from each other (Extended Data Fig. 11).

### Predictions on additional real data

We tested PEGSNet on all the subduction earthquakes (dip-slip mechanism within 40 km from the megathrust) with $M_w \geq 7$ that have occurred since January 2003, without considering aftershocks (Extended Data Fig. 10). Among them, the 2003 $M_w = 8.2$ Hokkaido earthquake is at the edge of PEGSNet's lower sensitivity limit of 8.3. For this event, PEGSNet estimates the final $M_w$ after about two minutes (Extended Data Fig. 10b), in agreement with what was previously observed on the test set for events with similar magnitude (Fig. 2a). However, given the expected lower accuracy and higher errors for this event, we consider these predictions less reliable. For lower-magnitude events, PEGSNet predictions converge toward the noise baseline of 6.5 or never exceed its lower sensitivity limit, confirming that PEGS from $M_w < 8.0$ earthquakes are essentially indistinguishable from noise (Extended Data Fig. 10c–f). Deep learning denoising techniques for coherent noise removal[50] might prove successful in enhancing PEGSNet performance and will be the subject of future work.

### Data availability

Waveforms and metadata used in this study were provided by the Global Seismograph Network[51], the New China Digital Seismograph Network[52], GEOSCOPE[53] (IPGP/EOST, 1982), the Japan Meteorological Agency Seismic Network (https://www.jma.go.jp/en/quake) and the National Research Institute for Earth Science and Disaster Resilience/F-net[54], all publicly available from the IRIS Data Management Center (https://ds.iris.edu/ds/nodes/dmc) and the F-net data centre (https://www.fnet.bosai.go.jp/top.php). Source data are provided with this paper.

### Code availability

The codes used to generate and analyse the data shown within this paper are available from the corresponding author upon request. PEGSNet is built and trained using PyTorch[55]. Waveforms were retrieved from IRIS with Obspy[56] and obspyDMT[57]. Figures were produced with the Generic Mapping Tool (GMT)[58] and matplotlib[59].

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

**Acknowledgements** This project has received funding from the European Research Council (ERC) under the European Union's Horizon 2020 research and innovation programme (grant agreement 949221). This work has been supported by the French government, through the UCAJEDI Investments in the Future project managed by the National Research Agency (ANR), ANR-15-IDEX-01. This work was granted access to the HPC resources of IDRIS under the allocations 2020-AD011012142, 2021-AP011012536 and 2021-A0101012314 made by GENCI. B.R.-L.'s work was supported by Institutional Support (LDRD) at Los Alamos (20200278ER). Numerical computations for the synthetic database of PEGS were performed on the S-CAPAD platform, IPGP, France. We thank M. Böse for providing the results of FinDer2 on Tohoku-Oki data.

**Author contributions** A.L. developed PEGSNet and produced all the results and figures presented here. Q.B. had the original idea and supervised the study. B.R.-L. designed the preliminary algorithm and provided the raw noise data. J.-P.A. provided expertise in modelling PEGS and realistic source time functions. K.J. provided the synthetic database of PEGS necessary to produce the training database. All the authors contributed to writing the manuscript.

**Competing interests** The authors declare no competing interests.

**Additional information**
**Correspondence and requests for materials** should be addressed to Andrea Licciardi.

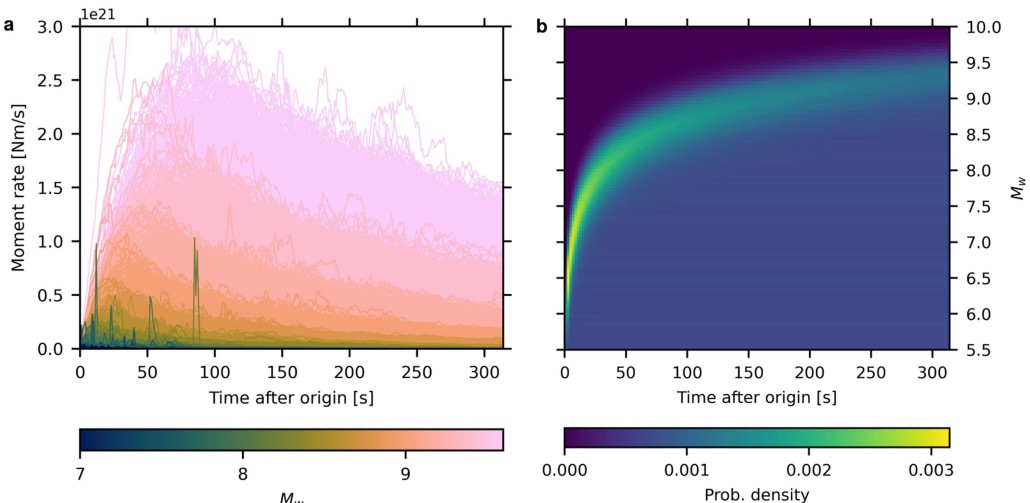

**Extended Data Fig. 1 | STF database. a**, STFs (moment rate) coloured by their final magnitude in the training database from a previous model[5] (see description in Methods). Only STFs for events with final $M_w$ between 7.0 and 9.5 are shown. **b**, The complete $M_w(t)$ distribution is used for labelling the database.

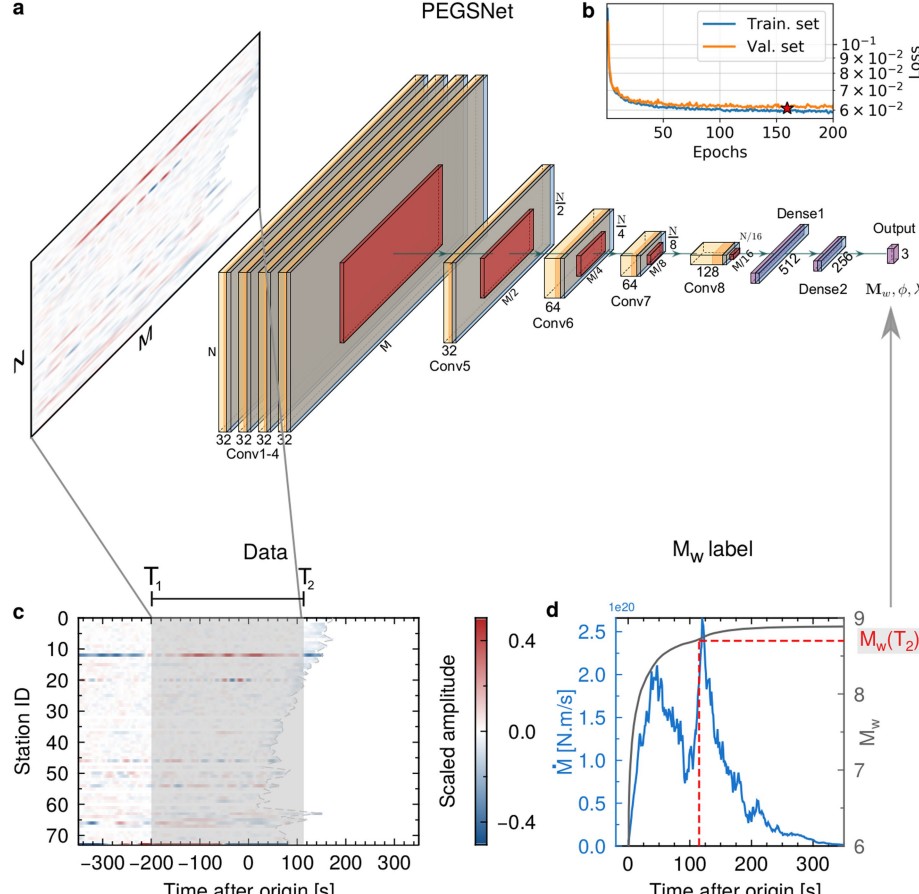

**Extended Data Fig. 2 | PEGSNet architecture and training strategy.**
**a**, The input data for one example consists of a three-channel image of shape
$M \times N$, where $M$ is the number of time samples and $N$ is the number of seismic
stations. Only the vertical component of the input data is displayed for
simplicity. Each convolutional block is composed of a convolutional layer
(yellow) with a ReLu activation (orange), a spatial dropout layer (light blue).
Max pooling layers (red) reduce each dimension of the input data by a factor of
two. The number of filters used in each convolutional layer is indicated for
clarity. The last convolutional block is connected to dense layers (purple)
(using a ReLu activation function), with dropout (light blue). The output layer
uses a tanh activation function to predict values of $M_w$, latitude ($\varphi$) and
longitude ($\lambda$). **b**, The value of the Huber loss is plotted as a function of epochs

for the training (Train.) and validation (Val.) sets. Each epoch corresponds to a
full pass over the training set in batches of size 512. The red star indicates the
epoch with the minimum value of the loss on the validation set. The
corresponding model is used for predictions on the test set and real data.
**c**, Data from one example from the training database (vertical component).
The grey shaded area corresponds to the input data for PEGSNet shown in **a**. $T_1$
and $T_2$ are the beginning and end of the selected input window. During training,
$T_1$ is selected at random and $T_2 = T_1 + 315$ s. **d**, Moment rate (blue) and $M_w(t)$ (dark
grey) for the selected event. Given the randomly selected value for $T_1$ for this
example, the corresponding label is $M_w(T_2)$, that is, at the end of the selected
window. This is compared with the predicted $M_w$ estimated by PEGSNet in **a** and
used for training.

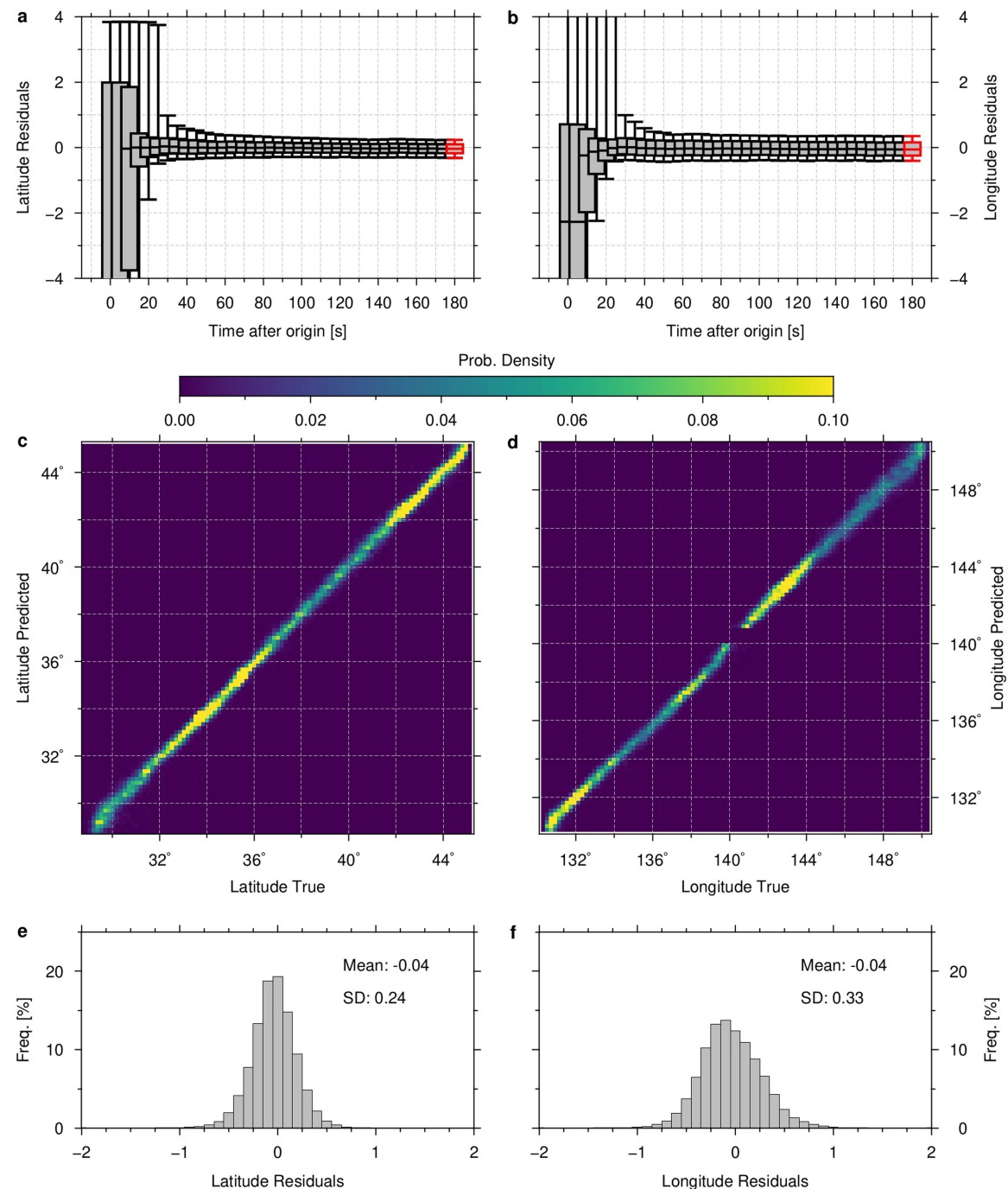

**Extended Data Fig. 3 | Results of predictions on the test set for location.**
**a**, **b**, Analysis of the residuals as a function of time (only the first 180 s from origin time are shown) for latitude (**a**) and longitude (**b**). Boxes correspond to the third to first (Q3–Q1) interquartile range, the black line within the boxes is the median and whiskers indicate the 5th and 95th percentiles. **c**, **d**, Density plot of true versus predicted latitude (**c**) and longitude (**d**) values at 180 s after origin (red boxes in **a**, **b**). **e**, **f**, Corresponding histogram of the residuals with reported mean and standard deviations for latitude (**e**) and longitude (**f**) at 180 s after origin.

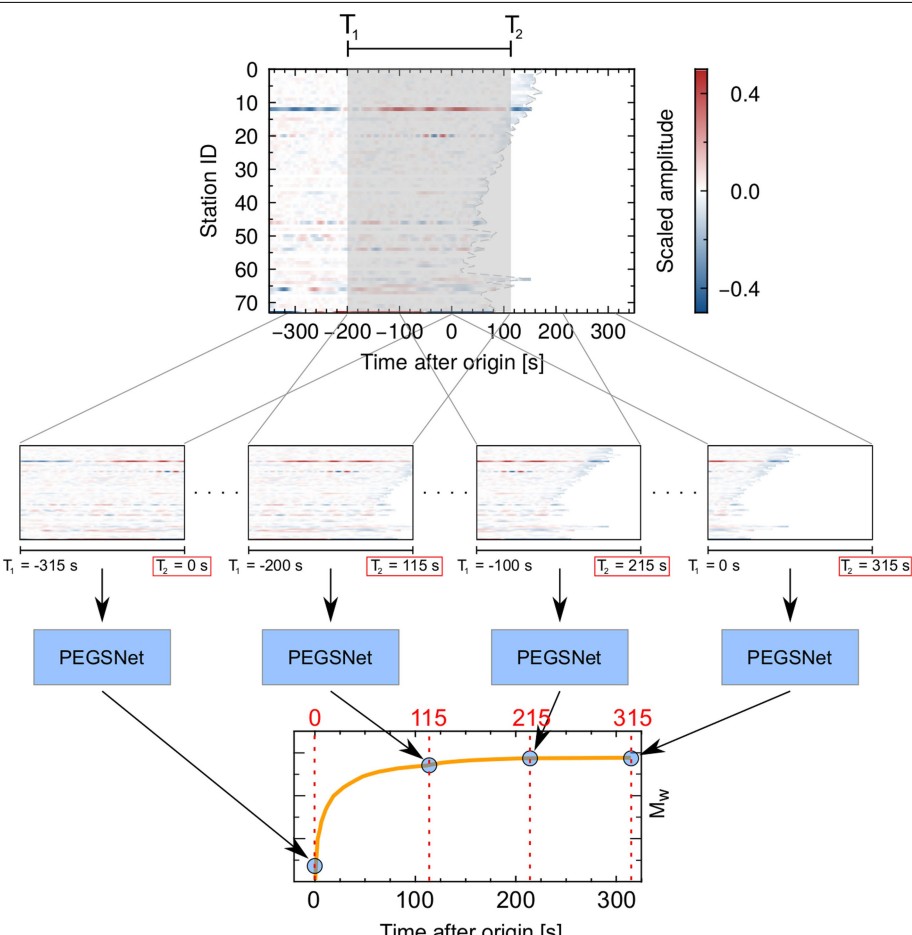

**Extended Data Fig. 4 | Testing procedure.** Each input example (top row) is parsed with a time window of length 315 s sliding with 1-s time step. At each time step, the data between $T_1$ and $T_2$ are passed to PEGSNet (middle row). For display purposes, only four specific time steps are shown here with their $T_1$ and $T_2$ indicated for clarity. The window with $T_1 = -200$ s and $T_2 = 115$ s corresponds to the grey shaded area in the top row. At each time step, PEGSNet makes a prediction of $M_w$ at the end of the input window, that is, $M_w(T_2)$ (blue circles in the bottom figure), to reconstruct the STF (yellow line) in a real-time scenario. Red dashed lines indicate the value of $T_2$ of each input window.

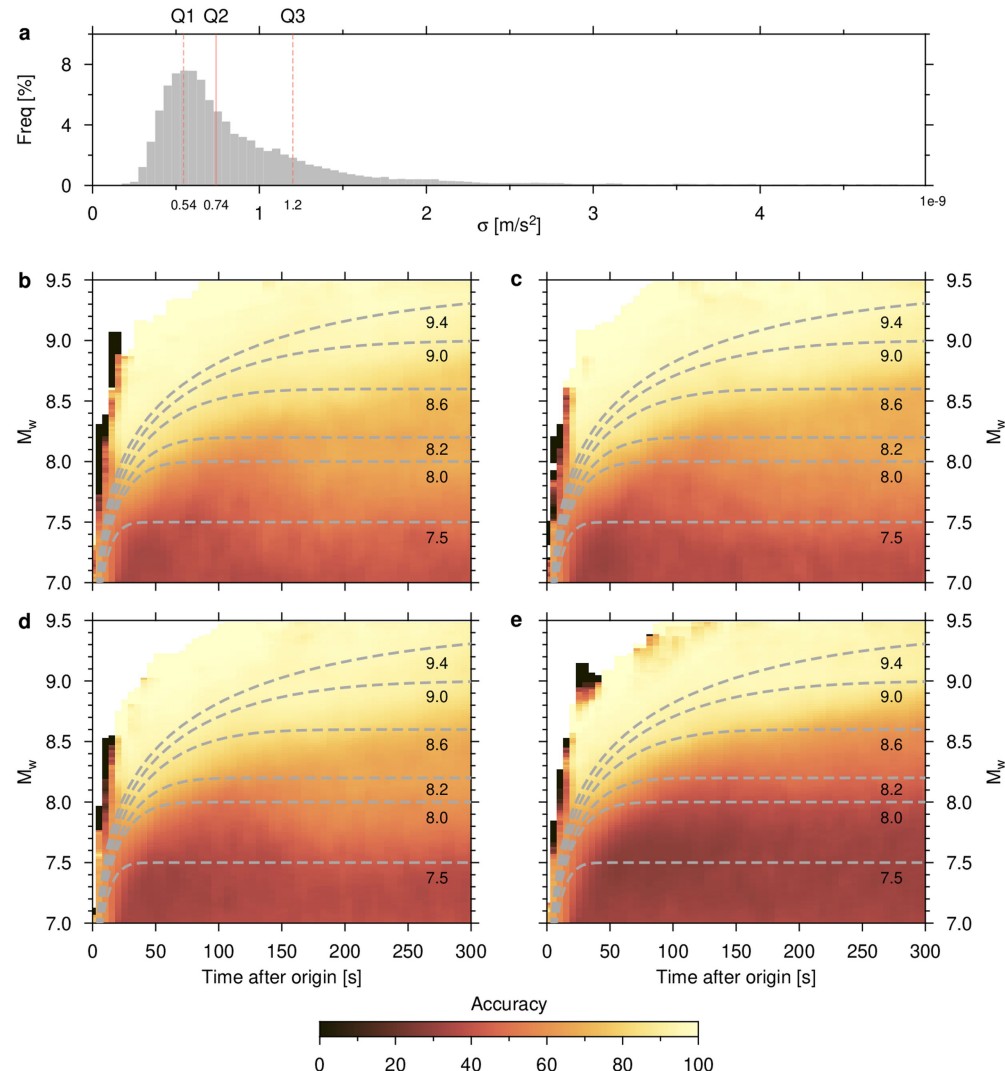

**Extended Data Fig. 5 | Effect of noise on PEGSNet predictions. a**, Frequency distribution of the mean standard deviation of the noise ($\sigma$) for the whole seismic network, in a pre-event time window of five minutes, for test set only. The solid red line is the median of the distribution (second quartile, Q2), dashed lines are the first (Q1) and third (Q3) quartiles. **b**–**e**, As in Fig. 2a, but computed on subsets of the test set for which $\sigma <$ Q1 (**b**), Q1 $< \sigma <$ Q2 (**c**), Q2 $< \sigma <$ Q3 (**d**), and $\sigma >$ Q3 (**e**). Note the high accuracy below 8.3 after 150 s from origin time for low-noise conditions (**b**). These maps give empirical limits on the noise levels that could potentially be useful to interpret the performance of our model in real time.

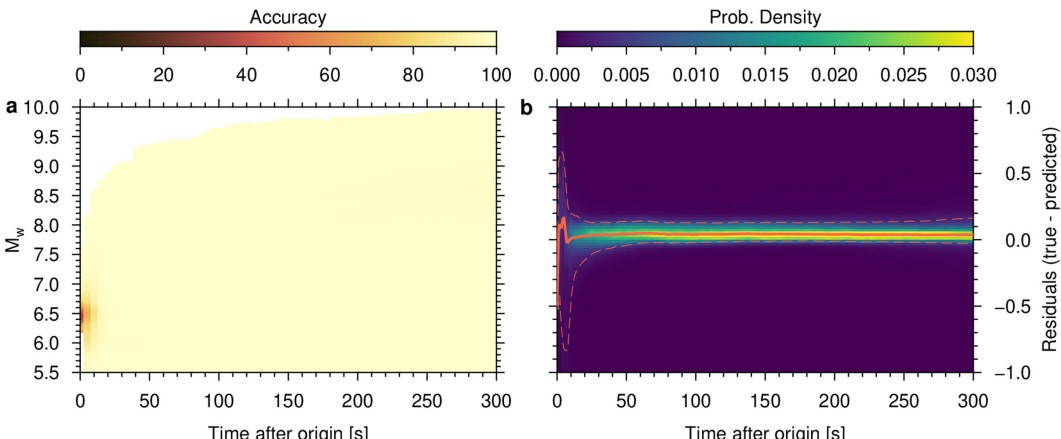

**Extended Data Fig. 6 | Results on noise-free test set. a**, As in Fig. 2a, but for a test set including only noise-free PEGS waveforms. **b**, Density plot of the residuals as a function of time for the same test set. The solid red line is the median, the dashed lines are the 5th and 95th percentiles of the distribution. Note that both in **a** and **b**, the predictions are obtained for a model that has been trained on a database of noise-free synthetics.

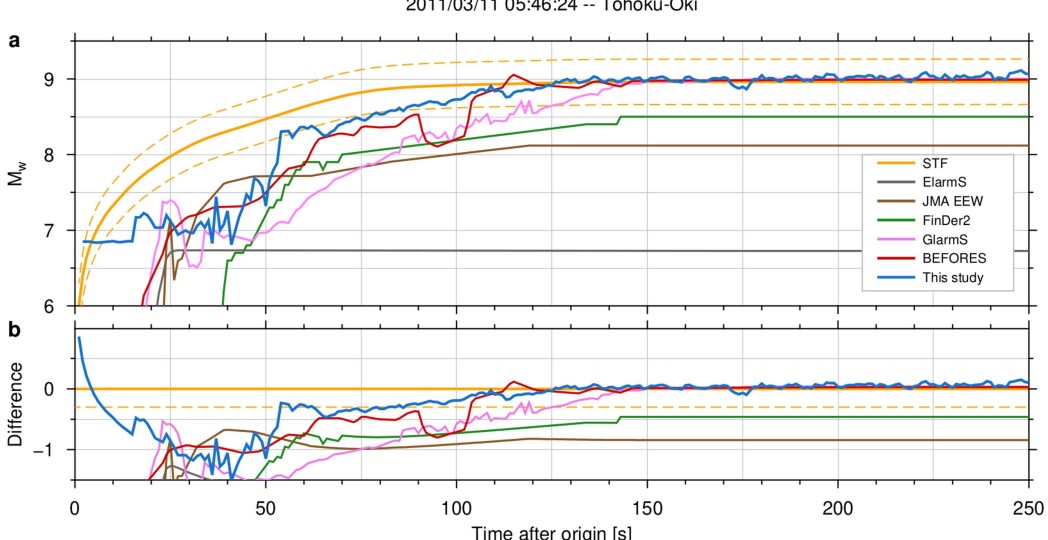

**Extended Data Fig. 7 | Comparison of PEGSNet with existing EEWS. a, b,** As in Fig. 3, but with two additional EEWS algorithms included. The results for Tohoku of finite-fault GNSS-based G-larmS[28] and seismic point-source ElarmS are taken from an earlier work[28]. In the main text, only the best-performing algorithm for each type (JMA, point-source; FinDer2, seismic finite-fault; and BEFORES, GNSS-based finite-fault) are shown.

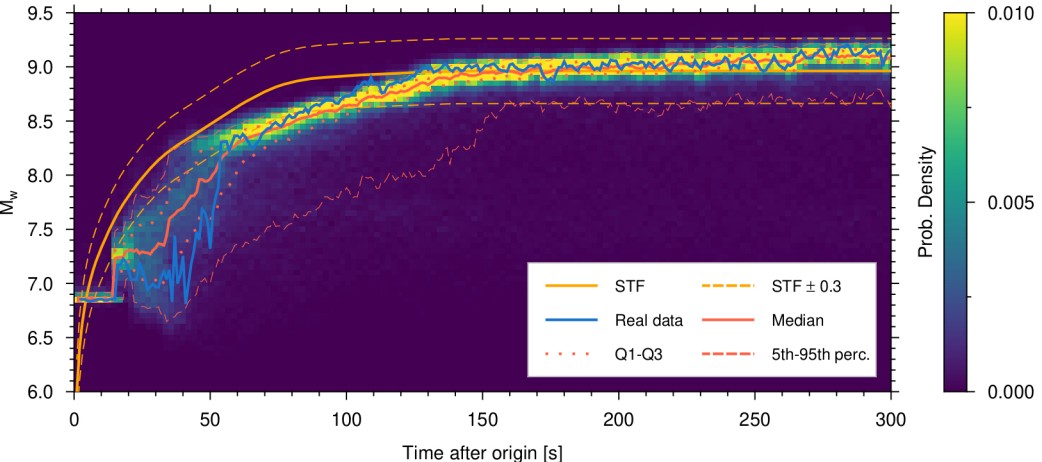

**Extended Data Fig. 8 | Synthetic test with Tohoku STF.** Density plot of the predictions on 1,000 examples obtained by combining different noise recordings (randomly extracted from the test set) with PEGS waveforms, from a synthetic source that mimics that of the Tohoku-Oki earthquake. The median, Q1–Q3 interquartile range and the 5th–95th percentiles of the distribution are reported with solid, dotted and dashed red lines, respectively. The synthetic Tohoku-like source is obtained from the 'true' STF (orange line, same as in Fig. 3 with $\pm 0.3$ $M_w$ indicated by dashed lines), the hypocentre location (Extended Data Fig. 10a), and the following values for the strike, dip and rake: 193.0°, 8.9° and 78.4° (USGS catalogue). The workflow described in Methods is used to combine synthetics and noise. PEGSNet prediction for real Tohoku data is reported for comparison (blue line, same as in Fig. 3). Even if it has been trained on synthetic data plus noise, PEGSNet is able to generalize well to real data (the blue line is within the expected variability of the predictions obtained on synthetics). In the first 50 s after origin, the variability of the predictions is high and strongly affected by noise (similar to Fig. 2c), whereas after 50 s, predictions have similar $M_w(t)$ values and therefore are more robust. Note that for the lowest noise conditions, faster PEGSNet response is virtually possible, as early as about 30 s after origin (as indicated by the 95th percentile of the distribution).

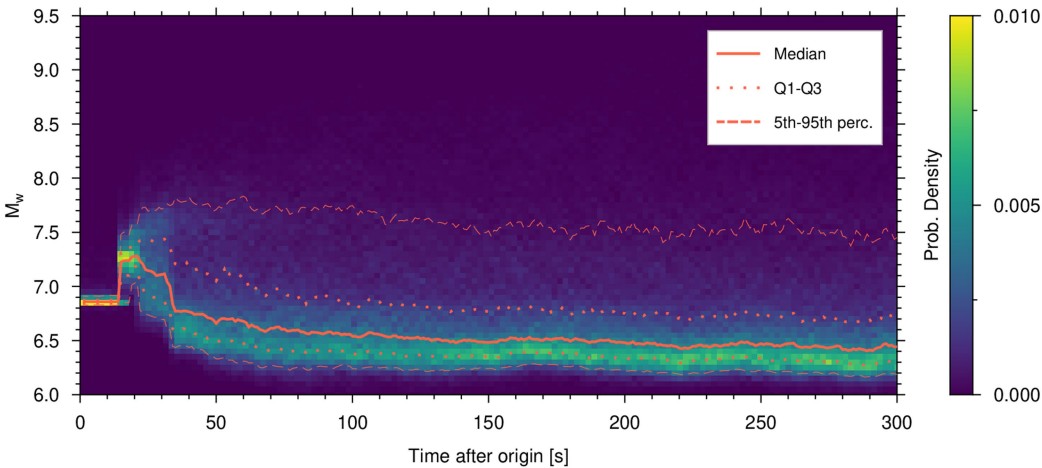

**Extended Data Fig. 9 | PEGSNet noise test.** Density plot of the predictions obtained for 1,000 real noise recordings (the same used in Extended Data Fig. 8) assuming P-wave arrivals (but no PEGS) as in the real data for the Tohoku-Oki earthquake. The median, Q1–Q3 interquartile range and the 5th–95th percentiles of the distribution are reported with solid, dotted and dashed red lines respectively. This test confirms that the results of Fig. 3 are indeed constrained by the data in the pre-P-wave time window (PEGS). The predicted $M_w(t)$ tends towards a constant value of 6.5, which is considered as the baseline value when no information can be extracted from PEGS.

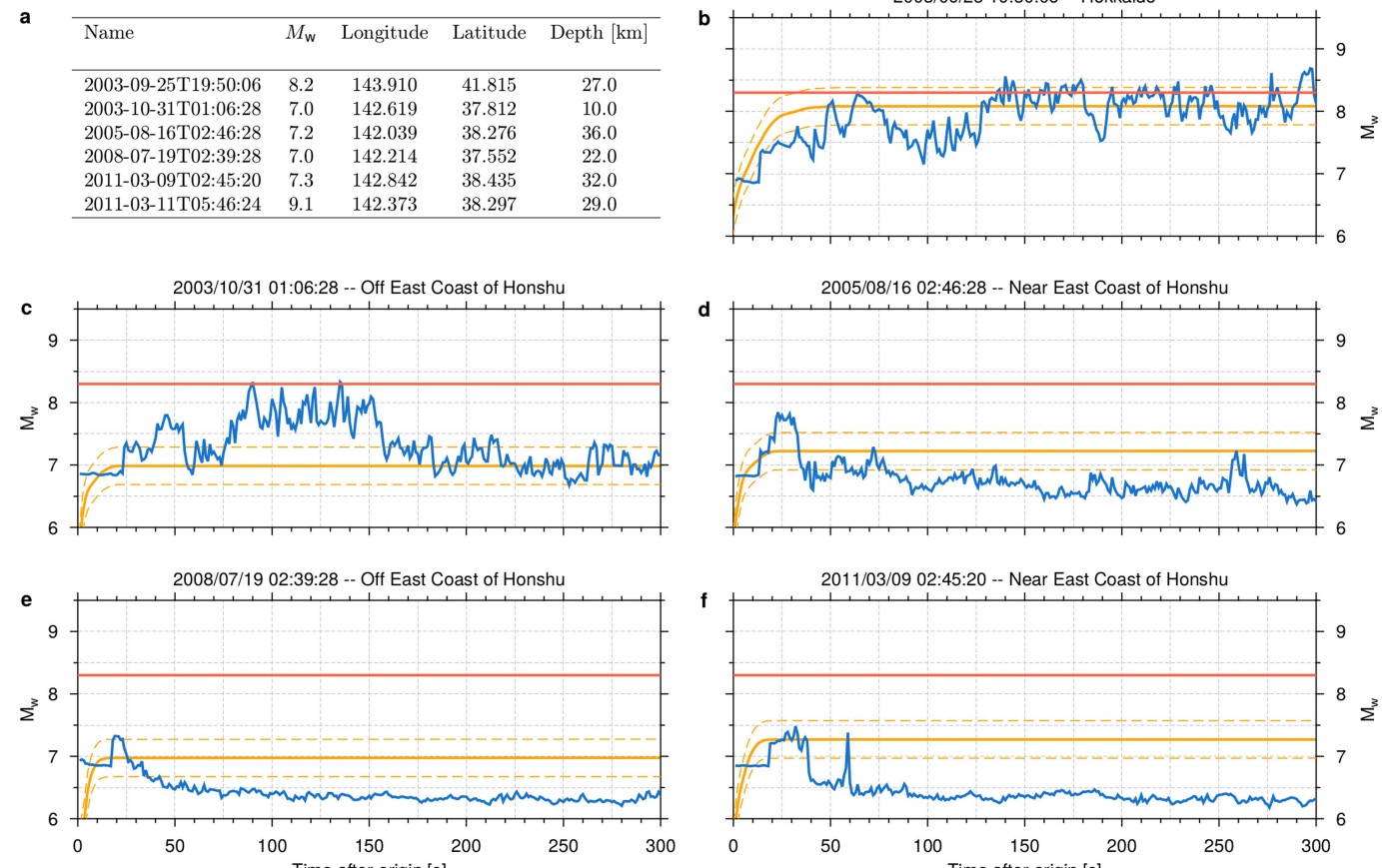

**Extended Data Fig. 10 | PEGSNet predictions on real data. a**, List of earthquakes used for the real data cases. All parameters are taken from the USGS catalogue. Earthquakes were selected according to the following criteria: $M_w \geq 7.0$, dip-slip focal mechanism and located within 40 km from the source locations used for training (green lines in Fig. 1). **b–f**, PEGSNet $M_w(t)$ predictions are indicated by the blue line. The integrated STFs (orange lines) are taken from the SCARDEC database with dashed lines representing ±0.3 magnitude units.

The red horizontal line marks the empirical lower limit of PEGSNet sensitivity (8.3). Note that for the 2003 $M_w$ 8.2 Hokkaido earthquake (**b**), this limit is crossed several times after 120 s from origin time. Out of four events with final $M_w$ below 8 (**c–f**), three (**d–f**) show a predicted $M_w(t)$ that is constant around the noise baseline (see Extended Data Fig. 9) indicating that no information can be extracted from the pre-P-wave time window.

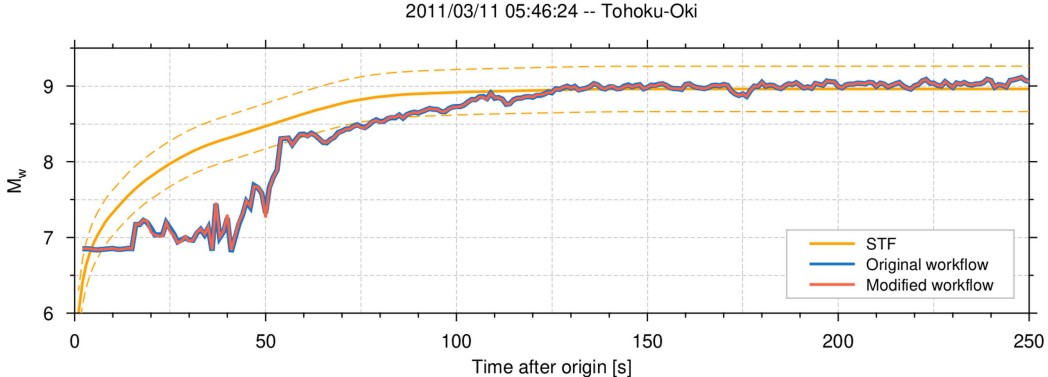

**Extended Data Fig. 11 | PEGSNet predictions for two data workflows.**
The predictions for the Tohoku-Oki data obtained through the original workflow described in Methods (blue line, same as in Fig. 3) are compared with those obtained with a modified workflow (red line). Instead of preprocessing the data once and then slice them into input windows for PEGSNet (original workflow), the data are preprocessed and fed to PEGSNet at each time step (modified workflow, see details in Methods) to simulate an online scenario. The results of two workflows show negligible differences. Note that the blue line is twice as thick as the red line. For reference, the orange lines indicate the true STF (solid) and ±0.3 magnitude units (dashed).