## [Peer Review File · Nature]

Manuscript Title: Instantaneous Tracking of Earthquake Growth With Elasto-Gravity Signals

Reviewer Comments & Author Rebuttals

Reviewer Reports on the Initial Version:

Referee #1:

Review of Licciardi et al. 2021-06-09579

In this review, I am addressing the review criteria requested by Nature. The criteria subheadings are indicated with ****Subheading****. This is followed by another section for my additional comments.

****Summary of the key results****

Licciardi et al. demonstrate how broadband seismometers can be used for rapidly estimating the magnitude of large ($M_w \geq 8$) earthquakes. As a type of Earthquake Early Warning (EEW), this approach relies on detecting Prompt Elasto-Gravity Signals (PEGs) – signals that (for large enough events) are detectable above the noise before the first arrival of seismic waves (P-waves).

****Originality and significance: if not novel, please include reference****

This is the first study to show how the PEGs can be used to estimate the evolving moment magnitude of large earthquakes. In that sense, it is novel. The detectability of PEGs for large events is now becoming well established (Vallée et al. 2017 and the subsequent studies referred to in this manuscript). The significance of this manuscript is that the use of PEGs for EEW seems to work as well as (if not better than) approaches based upon high rate GNSS (the previous best approach to avoid the magnitude saturation problem). Furthermore, the authors show that if seismic noise can further be reduced, the magnitude threshold for the detectability during the PEGs phase can be decreased – which would lead to a situation in which an even faster warning could be delivered (PEGs based EEW would then significantly outperform high rate GNSS). Seismometers are in situ measurements and therefore having an EEW method for large events that is not reliant on the GNSS data is undoubtedly very appealing to national earthquake monitoring agencies. This is because the accessibility of GNSS data should not be taken for granted (networks could be switched off at any time). The authors also refer to literature that suggests the use of other in-situ measurements such as gravity strainmeters that would further increase the effectiveness of using PEGs for EEW. Therefore, the impressive PEGs EEW presented by this paper could be even better in the near future.

****Data & methodology: validity of approach, quality of data, quality of presentation****

The seismic data and synthetic seismic data are well presented. The methods section adequately explains the treatment of noise in the training data (and makes sense).

As for the regression, I am not convinced that the choice of a CNN is really justified. The authors use

something like a “LeNet” type architecture, which is one of the simplest CNN architectures in common use. There is nothing wrong with simple. In fact, I am wondering if other perhaps simpler, non-neural network ML approaches could achieve equal performance on this regression problem. Accordingly, I strongly advise the authors to remove the “and Deep Learning” from the title. If the authors want to stick with the current CNN for the regression model, then I do not see a problem with this. The problem lies in that they have not convinced me that a deep-learning architecture is really necessary and therefore this deep-learning aspect could serve as a distraction in the paper. A related complaint is that the authors do not make any attempt to interpret the workings of the CNN. Some techniques exist for looking at converged CNNs layer by layer to see what types of data features are important for certain predictions. I think that some readers would be disappointed to see no analysis of the CNN, especially given that deep learning is given in the title. I think the easy fix here is to explain (somewhere between lines 68-74) that the CNN is not necessarily the best regression strategy and that the goal of this study is not to optimize the regression – rather it is to demonstrate EEW of large events with PEGs.

The regression is multi-output, solving for the earthquake location and moment release. It would be useful to see the performance of the location estimate with time in a revised version.

Of course, inclusion of realistic noise should be an important part of the analysis, but have the authors tried to run the CNN with a test-set of clean synthetic data? This could give an upper limit of model performance. Perhaps this could be included as an additional analysis of the methods section. I suspect that an upper limit of model performance on “perfect”, noise-free test-set data would depend on the richness of the training data examples.

Why do P-arrivals and later need to be zeroed out? This seems like it would be a pain to implement in an “online” version of this algorithm. What about keeping the scaling and clipping the post P-arrival data to the clip limits?

Lines 512-526: If this is running online (instantaneously), the filtering would have to take place on every 700s time step advanced. It seems that this reality is not considered in the chosen processing flow, which instead seems to filter the data (some of which is the “future” when the time window is earlier in the full time series). How can you filter the future? Does the processing of the data consider this practical need for iterative filtering? Or are the examples here all using an originally filtered trace that is subsequently sliced into sliding windows? If the current analysis does not consider this new filtering operation at every time step, then I suggest that the analysis is redone with a separate filter applied to each time 700 s window.

Figures are clear. One suggestion is that the noise level trace of figure S8 could also be included in figure 3.

Figure 1 caption refers to “dark green dots”. I do not see any – only dark-green lines (that might be dots that are too large and therefore merge into lines).

****Appropriate use of statistics and treatment of uncertainties****

To me it seems that there is one set of outputs for each time step test data set. Therefore, there are

no error statistics for a certain input. However, the authors do show a range of solutions in figure S7 for a case in which the same synthetic inputs are fed through the model with different noise windows. Therefore, I think S7 gives adequate information about the uncertainties of the model predictions.

Instead of only randomly zeroing out traces in the training-set, this could also be done in the evaluation of test-set data. This would result in a range of solutions. Another possibility would be to create many different CNNs with the same architecture, each with a subset of stations. This would be an ensemble of CNNs and would also give a range of solutions. It would be interesting to see if an ensemble approach could improve model performance. Accordingly, having features that are too similar to other features can degrade ML model performance (feature redundancy) and it is commonly advised to remove as much redundancy as possible.

All in all, a rigorous exploration of model uncertainties is not critical for this study, but some additional experimentation could improve the performance of the current model. I suggest that the authors play around and document any improvements in a revised version.

****Conclusions: robustness, validity, reliability****

Fault kinematic inversion approaches of GNSS data are criticized (e.g. line 62) but this is an unfair comparison as the authors use a different regression approach to get their M0 estimates. Also, could the GNSS data not also be put into a regression for epicentre and M0 using a similar ML approach?

I would like to see the authors better explain why their source time functions are not too simplistic or their synthetic model space for the training data is not too small. What about complex, multi-fault ruptures (e.g. Kaikoura), or debated role of splay fault activation during mainshock rupture periods (is all deformation occurring on the megathrust)? What about variations in rupture directivity? Data synthesis cannot be infinite, but also it might currently be too narrow in this current implementation. I think that authors should discuss this point about simplicity of training data in a revised version.

****Suggested improvements: experiments, data for possible revision****

Suggestions have been made in previous subsections.

****References: appropriate credit to previous work?****

Yes. This is appropriate.

****Clarity and context: lucidity of abstract/summary, appropriateness of abstract, introduction and conclusions****

I have small remarks to some of these portions and will list them in the additional subsection below.

Additional comments:

Line 14: "but are slower" - Too informal. Please provide numbers.

Line 18: "...zero time delay". Having read the paper, it is obvious what this sentence is getting at, but

it might seem to the new reader that the event is immediately detected as it is growing (which is practically impossible as all events start out small and therefore below the noise level). I suggest rephrasing this sentence to avoid any confusion.

Line 37: “small time window” – too informal. Please provide numbers.

Line 58: The word “timely” or “timeliness” appears frequently. I think a general reader would understand more easily the word “latency”. Consider changing relevant parts of the manuscript.

Line 86: How do the slip patterns look? Elliptical? Box-car?

Line 132: If results for location “rely on P arrival times”, then what do the model predictions for epicentre look like when no P’s have arrived? Because the model outputs M0, longitude, and latitude, right? When the earlier data inputs do not yet have any padded zeros (e.g. fig S4, T2=0s), the model is still going to give us a prediction of epicentre, right? Are epicentre predictions rubbish until the P’s arrive? Please show a plot of epicentre estimation as a function of time in a revised version.

Line 195: The authors of this manuscript also apply some arbitrary preprocessing to their data (e.g. filtering, clipping, etc). These steps are also increasing latency.

Line 477: Are longitude and latitude of the slip centroid?

Line 493: Is this 500,000 synthetic earthquakes or 500,000 fractions of a smaller amount of synthetic earthquakes? How are training samples shuffled? Does shuffling have a significant impact on the stability of the convergence?

Referee #2:

(By B. F. Chao.) I am requested, per the Editor, to comment on the seismological aspects of the work, considering my lacking of expertise in the area of deep learning.

This work on earthquake parameter determination based on prompt signal PEGS is significant, both in science and in practical implication. It represents a natural advancement following previous developments on the subject matter by a same associated group of authors. The speed-of-light transit of the gravity change signal is a physical truth. The modeling and calculation of PEGS as would be recorded by distant seismometers are confirmed in previous studies. Now it is time to make tangible experiments and implementation for harnessing PEGS in practical EEWs use. The present work demonstrates that deep learning is a feasible and effective approach for the latter, provided with the validity of its seismological contents. I have the following relatively minor comments:

In the synthetics random noises are added, but some important details about the noise levels are not made known until mentioned in the last part of the extended material. What are the levels of the added noise in terms of signal-to-noise ratio? On what ground are the levels chosen? What

frequency band? What are the effects of the noise for different levels (“will be the subject of future work”)? Some elaborations are in order.

It’d be informative to have more discussions on the large cancelling effect between the gravity and acceleration signals, especially in the early part of the sourcing. Its (negative) impacts on the signal amplitude as well as earliest underestimation of magnitude can be emphasized. What is the physics of this cancellation anyway? What if one can have a means to sense gravity change only (isolated from the acceleration)?

The term “ground truth” for the synthetic experiments is misleading, as there is actually no “ground” truth.

Author Rebuttals to Initial Comments:

REVIEWER 1

*****Summary of the key results*****

Licciardi et al. demonstrate how broadband seismometers can be used for rapidly estimating the magnitude of large ($M_w \geq 8$) earthquakes. As a type of Earthquake Early Warning (EEW), this approach relies on detecting Prompt Elasto-Gravity Signals (PEGs) – signals that (for large enough events) are detectable above the noise before the first arrival of seismic waves (P-waves).

*****Originality and significance: if not novel, please include reference*****

This is the first study to show how the PEGs can be used to estimate the evolving moment magnitude of large earthquakes. In that sense, it is novel. The detectability of PEGs for large events is now becoming well established (Vallée et al. 2017 and the subsequent studies referred to in this manuscript). The significance of this manuscript is that the use of PEGs for EEW seems to work as well as (if not better than) approaches based upon high rate GNSS (the previous best approach to avoid the magnitude saturation problem). Furthermore, the authors show that if seismic noise can further be reduced, the magnitude threshold for the detectability during the PEGs phase can be decreased – which would lead to a situation in which an even faster warning could be delivered (PEGs based EEW would then significantly outperform high rate GNSS). Seismometers are in situ measurements and therefore having an EEW method for large events that is not reliant on the GNSS data is undoubtedly very appealing to national earthquake monitoring agencies. This is because the accessibility of GNSS data should not be taken for granted (networks could be switched off at any time). The authors also refer to literature that suggests the use of other in-situ measurements such

as gravity strainmeters that would further increase the effectiveness of using PEGs for EEW. Therefore, the impressive PEGs EEW presented by this paper could be even better in the near future.

We thank Reviewer 1 for his encouragement and for the time and effort spent on our manuscript. In the following, we have addressed the Reviewer's comments with extensive additions and new figures.

R1.1: *As for the regression, I am not convinced that the choice of a CNN is really justified. The authors use something like a “LeNet” type architecture, which is one of the simplest CNN architectures in common use. There is nothing wrong with simple. In fact, I am wondering if other perhaps simpler, non-neural network ML approaches could achieve equal performance on this regression problem. Accordingly, I strongly advise the authors to remove the “and Deep Learning” from the title. If the authors want to stick with the current CNN for the regression model, then I do not see a problem with this. The problem lies in that they have not convinced me that a deep-learning architecture is really necessary and therefore this deep-learning aspect could serve as a distraction in the paper. A related complaint is that the authors do not make any attempt to interpret the workings of the CNN. Some techniques exist for looking at converged CNNs layer by layer to see what types of data features are important for certain predictions. I think that some readers would be disappointed to see no analysis of the CNN, especially given that deep learning is given in the title. I think the easy fix here is to explain (somewhere between lines 68-74) that the CNN is not necessarily the best regression strategy and that the goal of this study is not to optimize the regression – rather it is to demonstrate EEW of large events with PEGs.*

Indeed, our goal here was to demonstrate the feasibility of machine learning-based EEW with PEGs, and the simple CNN we chose may very well be improved upon. Following the Reviewer's suggestion, we made this clear in the main text (lines 113-115) and removed “deep learning” from the title (see also E2). We argue, however, that deep learning models allow the direct use of seismogram data (here we arranged the seismic data of the network in an image-like input) as opposed to classical machine learning models for which a set of features need to be built *a priori* from the input waveforms. We added one additional statement discussing this in Methods (lines 467-470).

R1.2: *The regression is multi-output, solving for the earthquake location and moment release. It would be useful to see the performance of the location estimate with time in a revised version.*

We have added two panels to Extended Data Figure 3 (panels a and b) in which we show the performance of the location estimate as a function of time on the test set. We note that after about 50 s from origin time the predictions show relatively constant errors. We also modified the text to mention this new piece of information (line 140).

R1.3: *Of course, inclusion of realistic noise should be an important part of the analysis, but have the authors tried to run the CNN with a test-set of clean synthetic data? This could give an upper limit of model performance. Perhaps this could be included as an additional analysis of the methods section. I suspect that an upper limit of model performance on “perfect”, noise-free test-set data would depend on the richness of the training data examples.*

This is definitely a valid point. However, the model has never seen noise-free synthetic data during training and therefore, testing the current model on noise-free data would not represent a true assessment of the model’s upper limit performance, in our opinion. Even if that was the case, this upper limit performance would be virtually impossible to achieve on real-world applications of our model where there is no such thing as perfect data. Nevertheless, testing on noise-free synthetics can reveal some inherent limitations of our model due to e.g. the seismic network geometry. In order to explore this aspect, we retrained the model on noise-free synthetics. The results are shown in the new Extended Data Figure 6. We followed the Reviewer’s comment and expanded the Method section to discuss this new additional analysis and briefly mention it in the main text (lines 163-165). Overall, this gives a limit on the fastest achievable predictions with our approach which is in the order of 10-15 s from origin time. Nevertheless, after this delay, the predictions are virtually perfect (high accuracy and error less than ± 0.1 Mw units) regardless of the magnitude. Therefore, the lower sensitivity limit of our approach depends on the noise level (as shown in Extended Data Figure 5), and it could be significantly lowered, depending on the overall noise conditions of the seismic network of interest.

R1.4: *Why do P-arrivals and later need to be zeroed out? This seems like it would be a pain to implement in an “online” version of this algorithm. What about keeping the scaling and clipping the post P-arrival data to the clip limits?*

We agree with the Reviewer that including P-waves in the analysis can facilitate the online implementation of our model. In addition, this would also increase the amount of information used by the model and possibly improve the performance of magnitude estimation. We have decided not to include data after the P-wave arrivals for two reasons. First, the main objective of this work is to assess PEGS potential for early warning and therefore excluding P-waves from the analysis gives a means to quantify the contribution of PEGS (and PEGS only) to the magnitude estimation problem. By excluding P-waves from the analysis, we provide indisputable evidence that PEGS can be used for early warning, as we stated in the manuscript (lines 219-221). The second reason is a more technical one. At the present time, our code for simulating PEGS does not accurately model P-waves. Because we rely on synthetic data for training, this limitation would introduce a strong bias when dealing with real data.

We are currently working on overcoming this limitation and including P-waves in a future version of the model. Given that amplitudes between PEGS and P-waves differ by orders of magnitude, we need a new strategy to deal with both signals for effective training. One possible solution is the one mentioned by the Reviewer (i.e. keep the clipping and scaling) but some information may be lost in the process. In any case, we believe that in the future combining PEGS data with additional observables (e.g. seismic waves, GNSS data) could prove beneficial for better early characterization of earthquake sources.

R1.5: *Lines 512-526: If this is running online (instantaneously), the filtering would have to take place on every 700s time step advanced. It seems that this reality is not considered in the chosen processing flow, which instead seems to filter the data (some of which is the “future” when the time window is earlier in the full time series). How can you filter the future? Does the processing of the data consider this practical need for iterative filtering? Or are the examples here all using an originally filtered trace that is subsequently sliced into sliding windows? If the current analysis does not consider this new filtering operation at every time step, then I suggest that the analysis is redone with a separate filter applied to each time 700*

s window.

The Reviewer is correct. In an online version of the model, all the pre-processing steps detailed in Methods (subsection **Preprocessing of real data**) need to be applied each time that new packets of data are streamed in. As noted by the Reviewer, the results presented in the manuscript do not take this into account: we preprocessed the data once and then sliced the input data in sliding windows. We refer to this as the original workflow (OW). In order to understand the consequences of this simplification on the results, we repeated the analysis for the Tohoku-Oki data with the approach suggested by the Reviewer: we apply the pre-processing steps detailed at lines 556-567 at each time step (but in step 1 we take 1h long traces ending at the current time step) and feed the corresponding data to PEGSNet. We refer to this as the new workflow (NW). It is instructive to first look at the difference in the data obtained with the OW and NW (Figure R1).

Figure R1: We plot the waveform obtained with OW and NW for station INCN at the first (a) and last (b) considered time steps. Note that for display purposes, OW is plotted with a line that is twice as thick as NW. (c) shows the mean difference between the waveforms obtained with the two procedures at each time step (the red vertical line indicates the P-wave arrival for this station). Although the overall difference is small (in the order of 10^{-12} m/s^2), the difference is larger at early times after origin.

In both OW and NW, the pre-processing is applied to one hour of data before the P-wave arrival time (OW) or the current time step (NW). The difference between the two versions of pre-processed data is smaller than 1% of the overall signal amplitude and becomes smaller as the current time step in NW approaches the P-wave arrival time in OW (Fig. R1c). This difference likely arises from both the demeaning operation and the instrument response removal (Steps 2 and 3 of the pre-processing). Thanks to the use of causal filtering operations, there is very little difference between the two workflows. We reprocessed all stations for the Tohoku-Oki event and observed similar behaviour. In any case, when the input data obtained with NW are fed to PEGSNet, the difference with the results obtained from input data preprocessed with OW is practically zero (Figure R2):

Figure R2: PEGSNet predictions on the Tohoku-Oki data obtained using the original workflow (OW) described in the manuscript and the new workflow (NW) where all the preprocessing steps are applied at each time step. Note that for display purposes, OW is plotted with a line that is twice as thick as NW. The “true” STF for Tohoku with ± 0.3 magnitude units is shown with solid and dashed red lines (same as in Fig. 3 of the manuscript).

Given the negligible effect on M_w estimation between the two workflows, we did not change the results presented in the manuscript.

For completeness, we note that, in this work, no attempts have been made to quantify the

latency introduced by the preprocessing operations on PEGSNet, as mentioned in the original version of the manuscript (lines 233-234). Our experience suggests that the instrument response correction will likely contribute to most of the latency, while filtering is usually faster. In addition, because we only use data before P-waves, the latency of the system is expected to be progressively reduced with time, as once a station has been reached by the P-wave there is no need to apply preprocessing for that station at subsequent time steps. We are currently experimenting with new procedures to optimize the operational performance of PEGSNet in real-time. However, we believe that this aspect goes beyond the scope of the presented work.

R1.6: *Figures are clear. One suggestion is that the noise level trace of figure S8 could also be included in figure 3.*

We have decided to keep Figure 3 as it is. We believe that introducing more content in this figure might distract the reader from the main conclusion of our work.

R1.7: *Figure 1 caption refers to “dark green dots”. I do not see any – only dark-green lines (that might be dots that are too large and therefore merge into lines).*

The Reviewer is correct: the lines are made of dots too large to be seen individually. We have changed the caption of Figure 1 to make this more clear. We thank the Reviewer for pointing this out.

R1.8: *To me it seems that there is one set of outputs for each time step test data set. Therefore, there are no error statistics for a certain input. However, the authors do show a range of solutions in figure S7 for a case in which the same synthetic inputs are fed through the model with different noise windows. Therefore, I think S7 gives adequate information about the uncertainties of the model predictions.*

In order to better constrain uncertainties on the model predictions, we repeated the analysis presented in the original Figure S7 on a larger ensemble of input data. Instead of using only

four time windows of noise, we combined the synthetic PEGS waveform of the Tohoku-like source with one thousand examples of noise recordings randomly extracted from the test set. This gives us a more robust quantification of uncertainties that are in line with our previous findings. The new Extended Data Figure 8 shows the results of this analysis. In addition, we used the same ensemble of noise recordings to repeat the experiment presented in original Figure S8, in which PEGSNet is called to predict $M_w(t)$ directly on noise waveforms but keeping the P-wave arrival times from the Tohoku earthquake (new Extended Data Figure 9). This allows us to quantify the uncertainties associated with the noise baseline. The likelihood of false positives, in this case, is practically zero considering the lower sensitivity limit of PEGSNet (about 8.3 / 8.4). Overall, we believe these two new (improved) figures add statistical significance to our previously presented results. We thank the Reviewer for the suggestions.

Instead of only randomly zeroing out traces in the training-set, this could also be done in the evaluation of test-set data. This would result in a range of solutions. Another possibility would be to create many different CNNs with the same architecture, each with a subset of stations. This would be an ensemble of CNNs and would also give a range of solutions. It would be interesting to see if an ensemble approach could improve model performance. Accordingly, having features that are too similar to other features can degrade ML model performance (feature redundancy) and it is commonly advised to remove as much redundancy as possible.

Regarding this point, we first would like to note that the random station muting is already applied on the test set. Therefore, Fig. 2 already takes into account the variability of the predictions associated with this over the whole test set. However, as pointed out by the Reviewer, the quantification of prediction uncertainties on one single example was somewhat overlooked in the original version of the manuscript. We believe that the new Extended Data Figure 8 mitigates this shortcoming. Nevertheless, we directly followed the Reviewer's suggestion and performed an additional test on Tohoku data. We obtained one thousand predictions of $M_w(t)$ using input data from Tohoku, but for each prediction, we muted four different (and randomly selected) stations (this conforms with the 5% muting strategy used during training). On top of that, we activated the dropout layers at inference time. This has been shown to provide an estimate of the posterior model uncertainties (Gal, & Ghahramani, 2016). From the retrieved ensemble of predictions, we estimate the median and the 5-95

percentile range. The results are compared with the original prediction (Fig.3a of the manuscript) and are shown in the following figure:

Figure R3: Uncertainties on PEGSNet predictions on the Tohoku-Oki data obtained by an ensemble approach (one thousand times), by combining Bayesian dropout and random station muting (4 stations). Blue lines are the median (solid) of the ensemble of the predictions and the 5-95th percentiles (dashed). The original prediction (Fig.3 of the manuscript) is drawn in red for comparison. The “true” STF for Tohoku with ± 0.3 magnitude units is shown with solid and dashed orange lines (same as in Fig. 3 of the manuscript).

The evolution of the uncertainties obtained for this ensemble test follows what we had observed for new Extended Data Figure 8: higher variability in the predictions in the first 50 seconds after origin, smaller errors after 50 s. Because the range of solutions obtained in this test is in some sense included in the uncertainties provided in the new Extended Data Figure 8, we decided not to add this figure in the manuscript considering the Editor’s guidelines on the number of Extended Data Figures.

All in all, a rigorous exploration of model uncertainties is not critical for this study, but some additional experimentation could improve the performance of the current model. I suggest that the authors play around and document any improvements in a revised version.

We believe that our new Extended Data Figures 8 and 9 address the main concerns of the Reviewer in terms of quantification of uncertainties as motivated above. We also performed an additional test on Tohoku data aimed at estimating uncertainties on a single set of input

but decided not to include it in the manuscript as justified above. Related to this comment, we believe that our test on noise free synthetics also adds significant value for the understanding of PEGSNet's performance (see comment R1.3).

R1.9: *Fault kinematic inversion approaches of GNSS data are criticized (e.g. line 62) but this is an unfair comparison as the authors use a different regression approach to get their M0 estimates. Also, could the GNSS data not also be put into a regression for epicentre and M0 using a similar ML approach?*

GNSS data can indeed be put into a similar regression for epicentre and M0 estimation using ML. In fact, this was done very recently (after the submission of our manuscript) by Lin et al. (2021). However, the approach needs to be significantly adapted to be based on GNSS rather than PEGS. First, modelling GNSS data requires a slip distribution while PEGS can be modelled accurately with a point-source approximation (see the response to comment R1.10). This makes a PEGS-based approach less sensitive to the details of earthquake ruptures, and therefore less susceptible to failure in the case of “exotic” earthquakes. Second, GNSS-based EEW is still fundamentally limited by latency because it is based on information carried at the P wave velocity, while PEGS travel at the speed of light. Unfortunately, a fair comparison between our PEGS-based approach and the GNSS-based ML approach developed in the aforementioned study is not possible because the authors ignored the time prior to the P-wave arrival time. Nevertheless, we believe a combined ML approach leveraging both GNSS and PEGS data, as well as seismic waves, could result in improved performance in the future. We added new text in the manuscript, updating the state of the art on GNSS-based EEWS and rephrasing the advantage of using PEGS over them.

R1.10: *I would like to see the authors better explain why their source time functions are not too simplistic or their synthetic model space for the training data is not too small. What about complex, multi-fault ruptures (e.g. Kaikoura), or debated role of splay fault activation during mainshock rupture periods (is all deformation occurring on the megathrust)? What about variations in rupture directivity? Data synthesis cannot be infinite, but also it might currently be too narrow in this current implementation. I think that authors should discuss this point about simplicity of training data in a revised version.*

One strength of the use of PEGS over GNSS for early warning applications is that PEGS are not sensitive to the details of the seismic rupture and therefore do not require modelling slip distributions. PEGS are sensitive to the magnitude, the focal mechanism and the source time function of the earthquake but, given the wavelength of the signal and the smoothness of the generated wavefield (Juhel et al., 2018), the spatial complexity of the rupture does not significantly affect the signal. For this reason, in this work, we only considered the most simplistic rupture spatial descriptions: point sources. The difference in PEGS predictions associated with the source finiteness was shown to be within the uncertainties of data during the 2011 Tohoku earthquake by Zhang et al. (2020) (see their Figure 3). Another observational evidence that this simplification is not problematic is the detection of PEGS generated by multiple events considering a simple point-source model (Vallée and Juhel, 2019). However, we considered a range of source time functions mimicking the historically observed variability of time-dependent rupture complexity (Meier et al., 2017). We added text in the manuscript reflecting this discussion (lines 41-46 and lines 90-96).

ADDITIONAL COMMENTS

R1.11: *Line 14: “but are slower” - Too informal. Please provide numbers.*

We have rewritten part of the abstract to address the Reviewer’s comments and to follow the Editor’s suggestions (see E5).

R1.12: *Line 18: “...zero time delay”. Having read the paper, it is obvious what this sentence is getting at, but it might seem to the new reader that the event is immediately detected as it is growing (which is practically impossible as all events start out small and therefore below the noise level). I suggest rephrasing this sentence to avoid any confusion.*

See our reply to the previous comment and E5.

R1.13: *Line 37: “small time window” – too informal. Please provide numbers.*

We have provided a range of values (from a few seconds to a few tens of seconds) as this

depends on the epicentral distance.

R1.14: *Line 58: The word “timely” or “timeliness” appears frequently. I think a general reader would understand more easily the word “latency”. Consider changing relevant parts of the manuscript.*

Agreed. The relevant parts of the manuscript have been modified.

R1.15: *Line 86: How do the slip patterns look? Elliptical? Box-car?*

We do not consider slip distributions, only point sources (see response to Comment R1.10).

R1.16: *Line 132: If results for location “rely on P arrival times”, then what do the model predictions for epicentre look like when no P’s have arrived? Because the model outputs M_0 , longitude, and latitude, right? When the earlier data inputs do not yet have any padded zeros (e.g. fig S4, $T_2=0s$), the model is still going to give us a prediction of epicentre, right? Are epicentre predictions rubbish until the P’s arrive? Please show a plot of epicentre estimation as a function of time in a revised version.*

We have included a plot of epicentre location as a function of time in new Extended Data Figure 3 (see also comment R1.2).

R1.17: *Line 195: The authors of this manuscript also apply some arbitrary preprocessing to their data (e.g. filtering, clipping, etc). These steps are also increasing latency.*

We have rephrased this part of the manuscript. We avoid commenting on latency and just highlight the advantages of our approach in terms of data selection (none is performed as the network is trained on all available data even if very noisy) and sensitivity (to the first order variations in the STF of the earthquake) (see also R1.9 and R1.10).

R1.18: *Line 477: Are longitude and latitude of the slip centroid?*

These are coordinates of the point source (see response to Comment R1.10).

R1.19: *Line 493: Is this 500,000 synthetic earthquakes or 500,000 fractions of a smaller amount of synthetic earthquakes?*

The full database is made of 500,000 synthetic earthquakes. We have made this explicit in the text (line 515).

How are training samples shuffled?

The training samples are selected at random from the entire database. We only make sure that training, validation and test set do not share the same empirical noise recordings and STF.

Does shuffling have a significant impact on the stability of the convergence?

Although we have never investigated this particular aspect explicitly, while tuning the model, we have used different shuffling orders and never noted significant differences related to that. In addition, because we are training with synthetic data, we implicitly impose that all the samples come from the same distribution which can explain why shuffling doesn't affect the learning performance of the model.

REVIEWER 2

This work on earthquake parameter determination based on prompt signal PEGS is significant, both in science and in practical implication. It represents a natural advancement following previous developments on the subject matter by a same associated group of authors. The speed-of-light transit of the gravity change signal is a physical truth. The modeling and calculation of PEGS as would be recorded by distant seismometers are confirmed in previous studies. Now it is time to make tangible experiments and implementation for harnessing PEGS in practical EEWS use. The present work demonstrates that deep learning is a feasible and effective approach for the latter, provided with the validity of its seismological contents.

We thank Reviewer 2 for the time and effort spent on our manuscript. In the following, we have addressed the Reviewer's comments, notably by extensively assessing the effect of noise level on the performance of our approach.

R2.1: *I have the following relatively minor comments: In the synthetics random noises are added, but some important details about the noise levels are not made known until mentioned in the last part of the extended material. What are the levels of the added noise in terms of signal-to-noise ratio? On what ground are the levels chosen? What frequency band? What are the effects of the noise for different levels (“will be the subject of future work”)? Some elaborations are in order.*

On the synthetics, we add real noise recorded at each individual station (coherently across the entire network, i.e. at the same time on each station). The procedure of noise selection and preprocessing is detailed in the Method section (lines 419-439). There, we also specify the frequency band of interest (2-30 mHz), which is the same used for synthetic PEGS waveforms and has proven to be effective for observation of PEGS on real data (Vallée and Juhel, 2019). One important aspect of our approach for the treatment of noise is that we do not make *a priori* assumptions on the levels and characteristics of the noise. On the contrary, we include all real noise conditions found in continuous seismic recordings in the specified period range (lines 419-421). This is because, in principle, we want our model to be able to generalize well under a broad range of noise conditions. We made this point more clear in

Methods (lines 427-431). Nevertheless, in the original version of the manuscript, we showed that the noise level affects prediction error and accuracy of our model, with low noise conditions resulting in smaller error and higher prediction accuracy. Motivated by the Reviewer's comment, we extended this analysis in the revised version adopting a more pragmatic approach. We first calculate the mean standard deviation of the noise for each example in the test and look at the whole distribution (new Extended Data Figure 5a). This empirical distribution depicts the variable noise conditions in the test set. We then calculate the Q1, Q2 and Q3 of this distribution and create four subsets of the test set based on these values. For each subset, we inspect the average residuals of the predictions as done in Figure 2b of the main text (new Extended Data Figure 5b-e). This analysis illustrates the effects of the noise for different levels as requested by the Reviewer. The analysis confirms what was observed in the original manuscript: the lower the noise level, the lower the prediction errors. In addition, this also gives empirical limits on the noise levels that could potentially be useful to interpret the performance of our model in real-time.

As for the Reviewer's comment on lines 214-215 of the original manuscript: these lines referred to possible future applications of deep learning denoising for enhancing PEGS detection. In this context, recent works have shown how noise can be effectively reduced through deep autoencoders (Zhu et al, 2019). We believe that in light of our new analysis (Extended Data Fig. 5) this technique can help in decreasing the lower limit of PEGSNet sensitivity, but we feel that a detailed discussion on this topic is beyond the scope of this work. Nevertheless, we have moved this part in Methods and added the relevant reference to clarify our statement.

R2.2: *It'd be informative to have more discussions on the large cancelling effect between the gravity and acceleration signals, especially in the early part of the sourcing. Its (negative) impacts on the signal amplitude as well as earliest underestimation of magnitude can be emphasized. What is the physics of this cancellation anyway? What if one can have a means to sense gravity change only (isolated from the acceleration)?*

As a ground-coupled sensor, a broadband seismometer records both the gravity change induced by the earthquake rupture and the inertial response of its surrounding medium to the gravity perturbation. This induced, inertial response exists because the gravity perturbation

acts as a secondary source of deformation inside the whole Earth. In an infinite medium, the direct and induced perturbations are theoretically identical and an inertial sensor deployed in such a medium would not record PEGS prior to the arrival of the P-waves. In a realistic earth model, the free surface breaks this symmetry and a perfect cancellation is no longer observed. The early instrumental response of a sensor prior to the arrival of the direct seismic waves has been described in detail in e.g. Vallée et al. (2017) and supplementary materials, Heaton (2017) and Juhel et al. (2019). Since these studies are properly referenced in our manuscript (line 16 and lines 30-32) and that the physics behind PEGS generation is not an original result of the presented work, we do not think it is necessary to expand the discussion on the cancellation effect observed in the early times of the rupture.

On another hand, sensors known as gravity strainmeters measure the differential acceleration between seismically isolated test masses and are then sensitive to the gravity change only. As we stated in the manuscript (lines 28-30), recent studies have shown the potential of these future generation sensors for a gravity-based EEW (Juhel et al, 2018). These sensors are still under development and are not under the scope of the presented work, which focuses on the assessment of an operational PEGS-based EEW relying on present instrumentation.

R2.3: *The term “ground truth” for the synthetic experiments is misleading, as there is actually no “ground” truth.*

Agreed. The relevant parts in the manuscript have been changed.

REFERENCES not cited in the main text

Gal, Z. & Ghahramani, Z. Dropout as a Bayesian approximation: representing model uncertainty in deep learning. *Proceedings of the 33rd International Conference on Machine Learning*, **48**, 1050-1059 (2016).

REFERENCES cited in the main text

Heaton, T. Correspondence: response of a gravimeter to an instantaneous step in gravity. *Nat. Commun.*, **8**, 1348 (2017).

Juhel, K. et al. Normal mode simulation of prompt elastogravity signals induced by an earthquake rupture. *Geophys. J. Int.*, **216**, 935-947 (2019).

Juhel, K. et al. Earthquake early warning using future generation gravity strainmeters. *J. Geophys. Res.: Solid Earth*, **123**, 10889-10902 (2018).

Lin, J. T., Melgar, D., Thomas, A. M., & Searcy, J. Early warning for great earthquakes from characterization of crustal deformation patterns with deep learning. *J. Geophys. Res. Solid Earth*, **126**(10), e2021JB022703 (2021).

Meier, M. A., Ampuero, J.-P. & Heaton, T.H. The hidden simplicity of subduction megathrust earthquakes. *Science*, **357**, 1277-1281 (2017).

Vallée, M., & Juhel, K. Multiple observations of the prompt elastogravity signals heralding direct seismic waves. *J. Geophys. Res.: Solid Earth*, **124**, 2970-2989 (2019).

Vallée, M. et al. Observations and modeling of the elastogravity signals preceding direct seismic waves, *Science*, **358**(6367), 1164-1168 (2017).

Zhang, S., Wang, R., Dahm, T., Zhou, S. & Heimann, S. Prompt elasto-gravity signals (PEGS) and their potential use in modern seismology. *Earth Planet. Sci. Lett.*, **536**, 116150, ISSN 0012-821X, (2020).

Zhu, W., Mousavi, S. M., & Beroza, G. C. Seismic Signal Denoising and Decomposition Using Deep Neural Networks. *IEEE Trans Geosci Remote Sens*, **57**(11), 9476-9488 (2019), doi: 10.1109/TGRS.2019.2926772.

Reviewer Reports on the First Revision:

Referee #2:

I am Reviewer 1 in the authors' responses document. I have read all responses to Reviewer 1 and Reviewer 2. I judge that most of the comments have been adequately addressed, either by changes to figures, changes to the text, or direct replies in the authors' responses document. Here are my additional comments:

****Improved exploration of model uncertainties****

The new Extended Data Figures 8 and 9 are helpful new inclusions for reviewers' questions about model performance from a statistical perspective (given natural noise variability). These figures are lacking legends (something I should have raised in the previous review). On ED Fig 8, it is not clear what is indicated by the orange/red dotted lines (which sit seem to be some type of error statistic on of the orange line). From the caption of ED Fig 9, it seems that this dotted line on ED Fig 8 is the Q1-Q3 interquartile range. Both figures need legends and ED Figure 8 needs a better caption.

Apart from legend issues, these figures show that PEGSNET model is clearly performing robustly given a reasonable range of input noise. These new figures (and the associated calculations) are a significant improvement on ED Figure 7 and 8 of the original manuscript.

Figure R3: These results correspond to the authors randomly muting the test data many times (with a similar proportion as done with the training) and assessing the model performance statistically. Authors are calling this an ensemble approach, but the model used is still a single neural network. My idea of an ensemble would be having different models (with smaller input tensors for different data subsets). Nevertheless, the test performed for making this figure is useful in showing the sensitivity of the result to missing stations. I think that this testing by the authors is a reasonable response, given my original query about ensembles. I agree that the testing done to produce new ED Figures 7 and 8 satisfies the testing of errors and that Figure R3 is a good candidate for omission if manuscript space is short.

****Some important information that should not only be in the response document****

The authors have very clearly addressed my comments about when the filtering is applied in the processing flow. The results show that the effect of when the filtering occurs is negligible. I suggest adding this information into the methods section and additionally adding figure R2 (if space allows). An alternative that I can suggest would be to remake all figures using the results gained from the new workflow (NW). I suspect that this detail about the filtering in processing flow would be something that other readers would spot. Therefore, given that the authors have addressed this concern with testing, it would be best to include this in the manuscript in some form.

Miscellaneous:

Line 140 of tracked changes manuscript: This should be "25 to 30" not "25/30 km".

Author Rebuttals to First Revision:

RESPONSE TO REVIEWER 1

We thank Reviewer 1 for the time and effort spent on our manuscript. Here we respond to any comments and suggestions in detail.

In the following, a code is given to each of the comments which are reported for clarity. This code is used to reference the addressed comment in the new version of the manuscript. Please refer to the attached annotated .pdf file where all the modifications are highlighted. Modifications of the original manuscript are indicated by a **bold-face** font in the new version.

REVIEWER 1

Improved exploration of model uncertainties

R1.1 *The new Extended Data Figures 8 and 9 are helpful new inclusions for reviewers' questions about model performance from a statistical perspective (given natural noise variability). These figures are lacking legends (something I should have raised in the previous review). On ED Fig 8, it is not clear what is indicated by the orange/red dotted lines (which sit seem to be some type of error statistic on of the orange line). From the caption of ED Fig 9, it seems that this dotted line on ED Fig 8 is the Q1-Q3 interquartile range. Both figures need legends and ED Figure 8 needs a better caption.*

We have added legends to both figures and clarified the caption of ED Figure 8.

Apart from legend issues, these figures show that PEGSNET model is clearly performing robustly given a reasonable range of input noise. These new figures (and the associated calculations) are a significant improvement on ED Figure 7 and 8 of the original manuscript.

We thank the Reviewer for motivating us to explore more in detail this aspect of the work.

Figure R3: These results correspond to the authors randomly muting the test data many times (with a similar proportion as done with the training) and assessing the model performance statistically. Authors are calling this an ensemble approach, but the model used

is still a single neural network. My idea of an ensemble would be having different models (with smaller input tensors for different data subsets). Nevertheless, the test performed for making this figure is useful in showing the sensitivity of the result to missing stations. I think that this testing by the authors is a reasonable response, given my original query about ensembles. I agree that the testing done to produce new ED Figures 7 and 8 satisfies the testing of errors and that Figure R3 is a good candidate for omission if manuscript space is short.

In light of the Reviewer's clarification on her/his original comment we acknowledge our broad use of the term "ensemble". As the Reviewer pointed, in our first reply, we focused on assessing uncertainties on a single prediction (for the Tohoku data) through a range of solutions (that we referred to as an ensemble) by changing the input data and not the structure of the model. Following the Editorial guidelines we will not include Figure R3 in the revised manuscript. We nevertheless thank the Reviewer for suggesting these additional tests that overall improved our understanding of PEGSNet performance.

Some important information that should not only be in the response document

R1.2 *The authors have very clearly addressed my comments about when the filtering is applied in the processing flow. The results show that the effect of when the filtering occurs is negligible. I suggest adding this information into the methods section and additionally adding figure R2 (if space allows). An alternative that I can suggest would be to remake all figures using the results gained from the new workflow (NW). I suspect that this detail about the filtering in processing flow would be something that other readers would spot. Therefore, given that the authors have addressed this concern with testing, it would be best to include this in the manuscript in some form.*

Following the Reviewer's comment and the suggestion made by the Editor, we have included Figure R2 of our previous reply as an Extended Data Figure in the new version of the manuscript. We have also added some text in Methods to discuss this important aspect.

Miscellaneous:

R1.3 *Line 140 of tracked changes manuscript: This should be “25 to 30” not “25/30 km”.*

This has been changed.